# Runx2 and Runx3 differentially regulate articular chondrocytes during surgically induced osteoarthritis development

Kosei Nagata[1], Hironori Hojo[2], Song Ho Chang[1], Hiroyuki Okada [1,2], Fumiko Yano[3], Ryota Chijimatsu[3], Yasunori Omata [1,3], Daisuke Mori[3], Yuma Makii[1], Manabu Kawata[1], Taizo Kaneko[1], Yasuhide Iwanaga[1], Hideki Nakamoto[1], Yuji Maenohara[1], Naohiro Tachibana[1], Hisatoshi Ishikura[1], Junya Higuchi[1], Yuki Taniguchi[1], Shinsuke Ohba[2,4], Ung-il Chung[4], Sakae Tanaka [1] & Taku Saito [1]✉

The Runt-related transcription factor (Runx) family plays various roles in the homeostasis of cartilage. Here, we examined the role of Runx2 and Runx3 for osteoarthritis development in vivo and in vitro. *Runx3*-knockout mice exhibited accelerated osteoarthritis following surgical induction, accompanied by decreased expression of lubricin and aggrecan. Meanwhile, *Runx2* conditional knockout mice showed biphasic phenotypes: heterozygous knockout inhibited osteoarthritis and decreased matrix metallopeptidase 13 (Mmp13) expression, while homozygous knockout of *Runx2* accelerated osteoarthritis and reduced type II collagen (Col2a1) expression. Comprehensive transcriptional analyses revealed lubricin and aggrecan as transcriptional target genes of Runx3, and indicated that Runx2 sustained *Col2a1* expression through an intron 6 enhancer when Sox9 was decreased. Intra-articular administration of Runx3 adenovirus ameliorated development of surgically induced osteoarthritis. Runx3 protects adult articular cartilage through extracellular matrix protein production under normal conditions, while Runx2 exerts both catabolic and anabolic effects under the inflammatory condition.

Osteoarthritis (OA) is the most prevalent form of the joint disease characterized by cartilage degeneration. Progression of OA eventually leads to disability in older adult patients, which imposes a great socioeconomic burden[1,2]. OA, a multifactorial disease, is caused by aging, inflammation, heavy weight, hard work, and other factors[3,4]. In the early stage of OA, cartilage degeneration is obvious in the superficial zone (SFZ) of articular cartilage, which has a specific role for joint lubricity by producing lubricin, encoded by the proteoglycan 4 (*Prg4*) gene[5,6]. Prg4-knockout (KO) mice display irregular surfaces at 2 months and subsequently exhibit non-inflammatory hyperplastic synovium[6]. The deeper zone (DZ) of articular cartilage consists of chondrocytes and extracellular matrix[7]. Chondrocytes maintain the equilibrium of extracellular matrix and cartilage homeostasis by producing cartilage matrix proteins such as type II collagen α1 (Col2a1) and proteoglycans including Aggrecan (Acan)[8,9]. Transcription of *Col2a1* is strongly induced by sex-determining region Y-box 9 (Sox9) through Sox9 consensus sites in introns 1 and 6[8,10–13]. Notably, SOX9 overexpression was shown to alleviate the progression of experimental OA[14].

[1]Sensory & Motor System Medicine, The University of Tokyo, 7-3-1 Hongo, Bunkyo-ku, Tokyo 113-8655, Japan. [2]Center for Disease Biology and Integrative Medicine, Graduate School of Medicine, The University of Tokyo, 7-3-1 Hongo, Bunkyo-ku, Tokyo 113-8655, Japan. [3]Bone and Cartilage Regenerative Medicine, The University of Tokyo, 7-3-1 Hongo, Bunkyo-ku, Tokyo 113-8655, Japan. [4]Department of Cell Biology, Institute of Biomedical Sciences, Nagasaki University, 1-7-1 Sakamoto, Nagasaki 852-8588, Japan. ✉e-mail: tasaitou-tky@umin.ac.jp

In contrast to such anabolic factors, matrix metalloproteinase 13 (Mmp13) plays essential roles in cartilage degeneration and OA development through the degradation of Col2a1[15–19]. MMP13 expression was low in normal and early degenerative cartilage, but strongly upregulated in late-stage OA specimens[20]. Increased MMP13 protein was observed in chondrocytes under inflammatory conditions, e.g., stimulation by interleukin-1-beta (IL-1β)[14]. Previous studies have revealed multiple transcription factors upstream of Mmp13 expression, including nuclear factor kappa-light-chain-enhancer of activated B cells (NF-κB)-associated factors[21], hypoxia-inducible factor 2-alpha (HIF-2α)[20–24], and Runt-related transcription factor 2 (Runx2)[17–19].

The Runx family (Runx1/Runx2/Runx3) plays central roles in the development and homeostasis of cartilage and bone via the DNA-recognition consensus motif TG(T/C)GGT[25–27]. Runx2 was identified as the master regulator of chondrocyte maturation and endochondral ossification[28,29]. Runx2 directly induces many kinds of chondrocyte hypertrophy-related genes, such as *Mmp13*, and positively regulates hypertrophic differentiation of chondrocytes and ossification[27,30]. In *Runx2* heterozygous-knockout mice, OA development was significantly reduced, and expression of MMP13 was decreased[31]. Similar results were observed in tamoxifen-inducible chondrocyte-specific Runx2-knockout mice, in which mRNA levels of *Runx2* in cartilage were decreased by about half[32]. In addition, chondrocyte-specific Runx2 overexpression led to post-traumatic OA progression[33]. These in vivo data and other experimental findings consistently indicate the catabolic roles of Runx2 in articular cartilage[34,35]. Previous studies have also implicated Runx2 in skeletal development. *Runx2* homozygous-knockout embryos display remarkable impairment of endochondral ossification[28,29]. Runx2 is regarded as indispensable during the late stages of endochondral ossification, hypertrophic differentiation of chondrocytes, and osteoblast generation[35]. However, Kimura et al. showed that Runx1 and Runx2 cooperatively induce Sox5 and Sox6, leading to Col2a1 induction during skeletal growth[36]. Liao et al. reported that chondrocyte-specific knockout of *Runx2* during skeletal growth leads to decreased expression of *Col2a1* and *Mmp13* in temporomandibular joints[37]. Nevertheless, these anabolic roles of Runx2 in chondrocytes have not been displayed in adult articular cartilage.

In contrast to the complexity of Runx2 effects, previous studies congruously suggest chondrogenic effects of Runx1[38]. Runx1 enhances cartilage matrix production, whereas Runx1 deficiency accelerates OA progression[38]. Although Runx3 has been shown to promote aggrecan expression[39] and is involved in regulation of chondrocyte proliferation and differentiation[40], roles of Runx3 in articular cartilage have not been shown by in vivo experiments. Taken together, we hypothesized that Runx2 and Runx3 may have important roles in the maintenance of adult articular cartilage.

In this study, we examined the roles of Runx2 and Runx3 in OA development using conditional-knockout mice. We employed two surgical OA models in which inflammatory responses are deeply involved, as well as an aging model exhibiting more natural disease progression, in *Col2a1-Cre*[ERT2] and *Prg4-Cre*[ERT2] mice. We further analyzed mechanisms underlying the regulation of articular chondrocytes by Runx2 and Runx3 using chromatin immunoprecipitation sequencing (ChIP-seq) and RNA sequencing (RNA-seq). We found that Runx3 contributes to the maintenance of articular cartilage through the induction of extracellular matrix, while Runx2 exerts both catabolic and anabolic effects in a biphasic manner under the inflammatory condition.

## Results

### Expression of Runx2 and Runx3 in articular cartilage during the development of surgically induced OA

To examine Runx2, Runx3, and Sox9 expression in articular cartilage during OA development, we prepared an 8-week-old mouse OA model by surgical resection of the medial collateral ligament and medial meniscus (medial model)[41]. Immunohistochemistry of normal cartilage showed that Runx2 protein was predominantly located in the DZ of preoperative articular cartilage, while Runx3 and Sox9 were found in all layers (Fig. 1a). After surgery, Runx3 and Sox9 started to decrease in surgically treated knee joints, accompanied by Col2a1. Runx2 was continuously detected in the DZ. Immunohistochemistry revealed enhanced expression of Mmp13, especially 2 weeks after surgery and predominantly in the DZ of articular cartilage at 4, 6, and 8 weeks after surgery (Fig. 1a). We additionally examined changes in Runx2, Runx3, and Sox9 expression in articular cartilage with aging. In the knee joints of WT mice, their expression decreased over time and was markedly weakened at 12 months of age (Fig. 1b).

We performed in vitro analyses to examine roles of Runx2 and Runx3 under inflammation using IL-1β, which is widely employed to model chondrocyte degradation[14,42–44]. Primary chondrocytes[45] were exposed to IL-1β at different concentrations for 24 h (Fig. 1c); or 1 ng/mL IL-1β for 0–48 h (Fig. 1d). *Runx3* was downregulated by IL-1β exposure in dose- (Fig. 1c) and time-dependent manners (Fig. 1d), but levels of *Runx2* mRNA were unchanged following IL-1β exposure (Fig. 1c, d). Conversely, exposure to IL-1β elicited dose- and time-dependent upregulation of *Mmp13* and suppression of *Sox9*, followed by a decrease of *Col2a1* (Fig. 1c, d). These results indicate that exposure of in vitro chondrocyte to 1 ng/mL IL-1β for 24 h reproduces similar expression patterns of marker genes and proteins observed in vivo surgical OA models, including stable expression of *Runx2*, reduced *Runx3, Sox9,* and *Col2a1*, and increased *Mmp13*.

### Runx3 knockout accelerated OA development through Prg4 and Acan suppression

To reveal the roles of Runx3 in whole or SFZ articular cartilage after skeletal growth, we compared *Runx3*[fl/fl] littermates[46] (R3-Cntl) with *Col2a1-Cre*[ERT2]; *Runx3*[fl/fl] (R3-cKO[ER])[47] and *Prg4-Cre*[ERT2]; *Runx3*[fl/fl] (R3-pKO[ER])[48] mice. We prepared three distinct OA models (Supplementary Fig. 1a–c). As a result of the medial model[41], OA development was significantly accelerated in R3-cKO[ER] knee joints compared with R3-Cntl joints (Fig. 2a). In the articular cartilage of sham-operated R3-cKO[ER] joints, Runx3-positive cells decreased by 70% (Fig. 2b). Positive areas of Acan and Prg4 decreased with Runx3 knockout (Fig. 2b). Similarly, OA development was significantly accelerated in R3-pKO[ER] joints (Fig. 2c). In the articular cartilage of sham-operated R3-pKO[ER] joints, Runx3-positive cells decreased by 40% and the positive area of Prg4 was decreased (Fig. 2d). In addition, 8 weeks after destabilizing the medial meniscus (DMM)[49] as a minor injury-induced OA model, OA development was also significantly accelerated in 24-week-old R3-pKO[ER] mice compared with R3-Cntl mice (Supplementary Fig. 2). In the aging model (Supplementary Fig. 1c), OA development was unchanged in R3-cKO[ER] mice compared with R3-Cntl mice (Fig. 2e, f). In the articular cartilage of 18-month-old joints, the rates of Runx3-positive cells were not different for either genotype (Fig. 2f), probably because Runx3 expression decreased with aging, as shown in Fig. 1b. Expression of the marker proteins in both mice was similar as well (Fig. 2f). By *Runx3* knockout in the whole articular cartilage, chondrocyte apoptosis was not significantly changed in the sham side, the surgical OA model, or with aging, as well as by *Runx3* knockout in the SFZ articular cartilage of the sham side (Supplementary Fig. 3a–d).

We then examined expression of OA-associated genes in SFZ and DZ cells from Runx3-knockout and control mice. To enhance knockout efficiency, we prepared *Runx3*[fl/fl] (R3-Cntl) and *Col2a1-Cre; Runx3*[fl/fl] (R3-cKO) mice, which displayed normal skeletal development (Supplementary Fig. 4a–g). *Prg4* was decreased in R3-cKO SFZ cells and *Acan* was also decreased in R3-cKO DZ cells (Fig. 2g). Other genes were not changed in SFZ cells or DZ cells (Fig. 2g). We then prepared SFZ and DZ cells from R3-pKO[ER] and R3-Cntl littermates, and treated them with water-soluble tamoxifen. Only *Prg4* was decreased in Runx3-deficient SFZ cells (Supplementary Fig. 5). We further performed Runx3

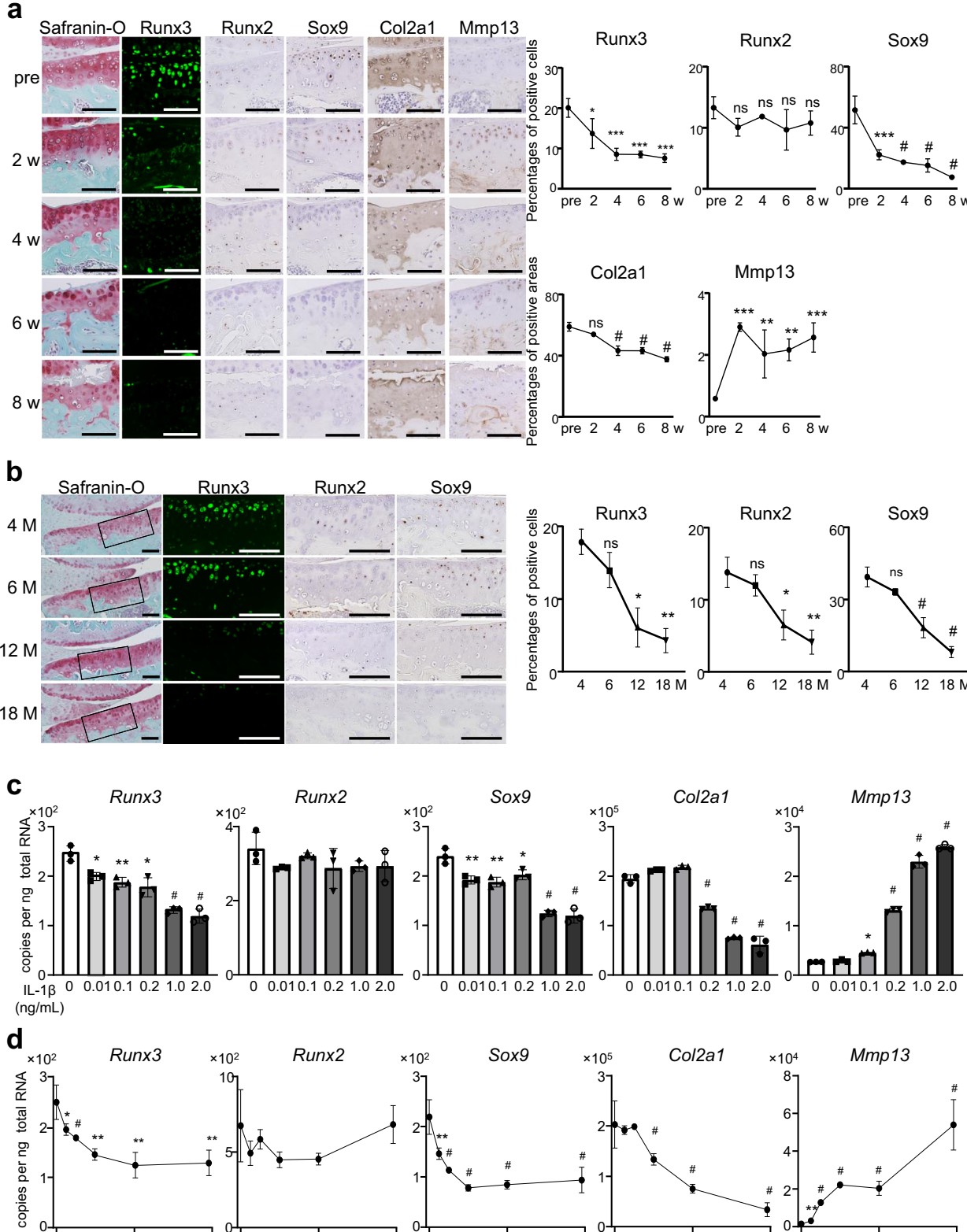

**Fig. 1 | Expression of Runx3 and Runx2, during osteoarthritis (OA) development. a** Immunofluorescence of chondrocyte markers during mouse OA development following resection of the medial meniscus and medial collateral ligament. Scale bars, 100 μm. Percentages of immunofluorescence-positive cells or areas in the Safranin-O-positive area are shown in right panels. **b** Immunohistochemistry of Runx3, Runx2, and Sox9 in knee joints of 4-, 6-, 12-, and 18-month-old mice. Scale bars, 100 μm. **c**, **d** mRNA levels of marker genes in mouse primary chondrocytes exposed to interleukin-1-beta (IL-1β) at concentrations ranging from 0.01 to 2 ng/mL for 24 h (**c**), or 1 ng/mL IL-1β for 0 – 48 h (**d**). Data are expressed as mean ± standard deviation (SD) of three biologically independent mice (**a**, **b**) or three experiments (**c**, **d**). *$P < 0.05$, **$P < 0.01$, ***$P < 0.001$, #$P < 0.0001$; ns, not significant; ordinary one-way ANOVA with Dunnett's multiple comparisons test (**a**, **b**), one-way ANOVA test with Turkey's post hoc test (**c**, **d**).

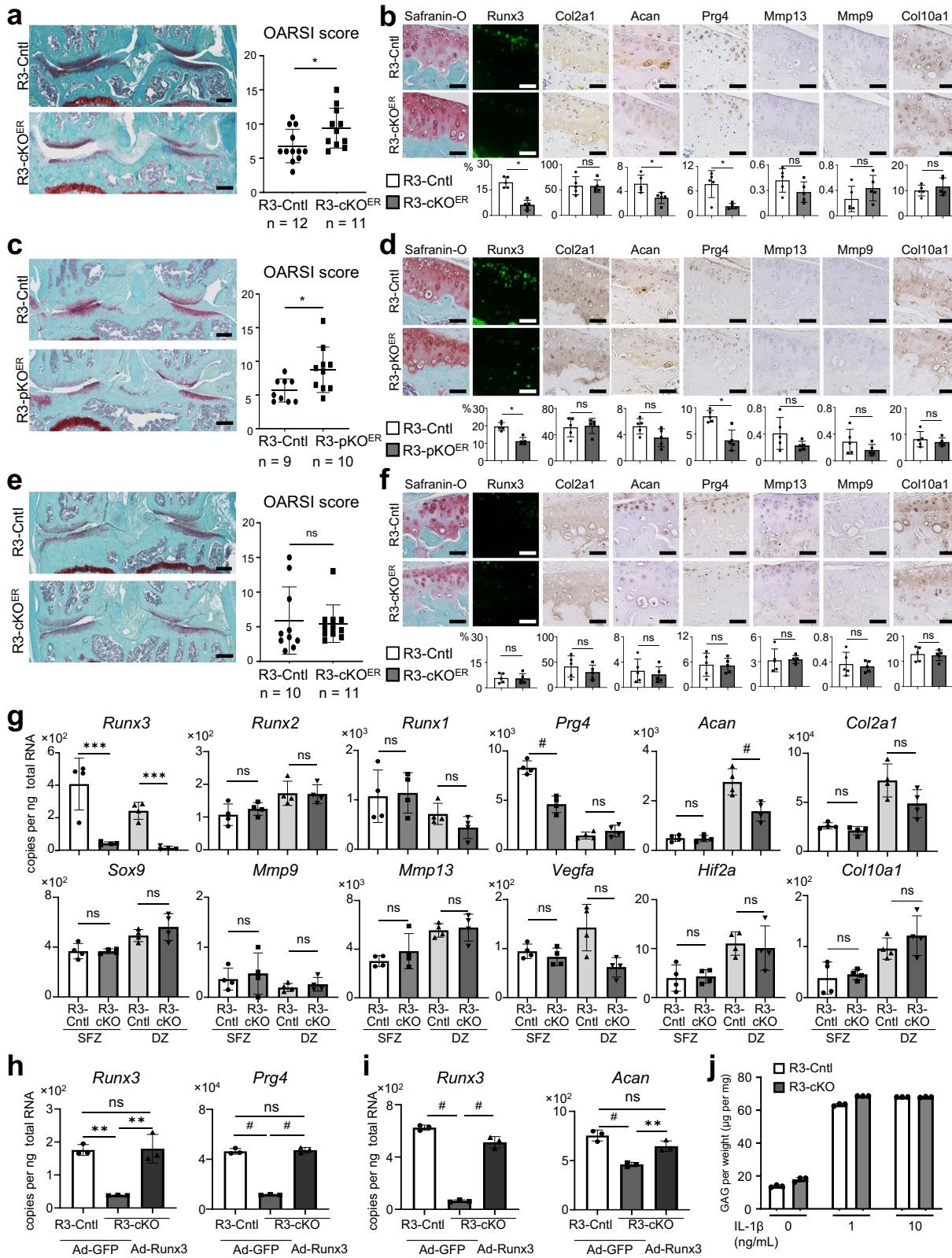

overexpression in Runx3-knockout cells using adenoviral transduction. Downregulation of *Prg4* in R3-cKO SFZ cells and *Acan* in R3-cKO DZ cells was recovered to control levels by Runx3 overexpression (Fig. 2h, i). Additionally, we performed dimethylmethylene blue (DMMB) assays[50] to investigate the catabolic effect of Runx3. Runx3 knockout did not significantly change the released glycosaminoglycan (GAG) content (Fig. 2j).

## Development of surgically induced OA was inhibited by heterozygous *Runx2* knockout, but accelerated by homozygous knockout

To investigate Runx2 loss-of-function in articular cartilage after skeletal growth in vivo, we generated a mutant mouse strain expressing a floxed *Runx2* allele (Supplementary Fig. 6a). Alizarin red and Alcian blue staining of skeletal preparations was performed in E18.5 embryos.

**Fig. 2 | Development of OA in cartilage-specific *Runx3*-knockout mice.** Development of OA in *Runx3*^fl/fl^ (R3-Cntl) and *Col2a1-Cre*^ERT2^; *Runx3*^fl/fl^ (R3-cKO^ER^) mice (**a**) and in R3-Cntl and *Prg4-Cre*^ERT2^; *Runx3*^fl/fl^ (R3-pKO^ER^) mice (**c**) by the medial model, and in R3-Cntl and R3-cKO^ER^ mice by the aging model (**e**). Scale bars, 200 μm (**a**, **c**, **e**). Immunofluorescence of marker proteins in sham-operated knee joints of 16-week-old R3-Cntl and R3-cKO^ER^ mice (**b**) and 16-week-old R3-Cntl and R3-pKO^ER^ mice (**d**), into which tamoxifen was injected at 7 weeks of age. Immunofluorescence of marker proteins in knee joints of 18-month-old R3-Cntl and R3-cKO^ER^ mice (**f**), into which tamoxifen was injected at 2, 6, and 12 months of age. Scale bars, 50 μm (**b**, **d**, **f**). Rates of Runx3-positive cells and the 3,3′-diaminobenzidine (DAB)-positive area in the Safranin-O-positive area are shown in the lower panels **b**, **d**. **f**. **g** mRNA levels of marker genes in superficial zone (SFZ) and DZ chondrocytes obtained from R3-Cntl and *Col2a1-Cre*; *Runx3*^fl/fl^ (R3-cKO) mice. **h**, **i** mRNA levels of *Runx3* and *Prg4* or *Acan* in the R3-Cntl and R3-cKO SFZ chondrocytes (**h**) or DZ chondrocytes (**i**) overexpressing GFP (Ad-GFP) or Runx3 (Ad-Runx3) by adenoviral induction at a multiplicity of infection (MOI) of 20. **j** Dimethylmethylene blue (DMMB) assays. Amount of proteoglycan released into culture medium from R3-Cntl and R3-cKO mouse femoral heads cultured with or without IL-1β (1, 10 ng per mL) for 3 days. Data are expressed as means ± SD of biologically dependent mice (*n* = 5 for **b**, **d**, **f**, *n* = 4 for **g**, and *n* = 3 for **h**–**i**). *$P < 0.05$, **$P < 0.01$, ***$P < 0.001$, #$P < 0.0001$, n.s., not significant; ordinary one-way ANOVA test with Turkey's post hoc test (**g**, **h**, **i**), and two-tailed Mann–Whitney *U* test (**a**–**f**, **j**).

*CAG-Cre; Runx2*^fl/fl^ mice exhibited a complete absence of mineralized matrix in developing skeletal bones, similar to global knockout mice (Supplementary Fig. 6b)[28,29]. Hypoplastic clavicles and open fontanelles were observed in *CAG-Cre; Runx2*^fl/+^ mice, which indicates a cleidocranial dysplasia phenotype, as reported in general heterozygous-knockout mice (Supplementary Fig. 6b)[19].

We next mated *Runx2*^fl/fl^ mice with *Col2a1-Cre*^ERT2^ mice[47] and generated *Runx2*^fl/+^ (R2-Hetero Cntl), *Col2a1-Cre*^ERT2^; *Runx2*^fl/+^ (R2-Hetero cKO), *Runx2*^fl/fl^ (R2-Homo Cntl), and *Col2a1-Cre*^ERT2^; *Runx2*^fl/fl^ (R2-Homo cKO) male littermates. In the medial model[41], OA development was significantly inhibited in knee joints of R2-Hetero cKO mice and accelerated in R2-Homo cKO mice compared with their respective controls (Fig. 3a). In the DMM model[49], OA development was significantly inhibited in R2-Hetero cKO knee joints, similar to previous reports[31,32], but accelerated in R2-Homo cKO mice compared with R2-Cntl joints (Fig. 3a).

Immunohistochemistry showed that Runx2-positive cells were reduced by approximately 40% in R2-Hetero cKO cartilage and 80% in R2-Homo cKO cartilage of sham and OA joints (Fig. 3b, c), respectively. Protein levels of Sox9 and Col2a1 were unchanged in the sham joints of both genotypes (Fig. 3b), but Col2a1 was significantly suppressed in OA joints of R2-Homo cKO mice (Fig. 3c). Mmp13 was significantly reduced both in sham and OA joints of R2-Hetero and R2-Homo cKO mice (Fig. 3b, c). In the aging model, we further compared OA development between R2-Homo Cntl and R2-Homo cKO mice. At 18 months of age, OA development with aging was unchanged in R2-Homo cKO mice compared with R2-Cntl mice, as well as immunohistochemistry of Runx2, Sox9, Col2a1, or Mmp13 (Supplementary Fig. 7a, b).

To examine chondrocyte apoptosis, we performed TdT-mediated dUTP nick-end labeling (TUNEL) staining of medial and DMM model knee joints. TUNEL staining revealed upregulated chondrocyte apoptosis in the sham joints of R2-Homo KO mice (Supplementary Fig. 8a). Cell numbers in articular cartilage were decreased in R2-Homo cKO mice (Supplementary Fig. 8a). To examine short-term effects of Runx2 knockout, we injected tamoxifen into 7-week-old male littermate mice of the four genotypes daily for 5 days, and sacrificed mice 1 week after injection without any surgery. Chondrocyte apoptosis was enhanced in the articular cartilage of R2-Homo cKO mice with decreased cell numbers (Supplementary Fig. 8b).

### Col2a1 expression was suppressed in homozygous *Runx2*-knockout chondrocytes under inflammation

Surgical induction differently altered OA development in R2-Hetero and Homo cKO mice, but had no effect on aging-model mice, indicating that Runx2 may be involved in cartilage degeneration by severe inflammation rather than aging with mild inflammation. To reveal the roles of Runx2 in chondrocytes under inflammation, we performed ex vivo and in vitro analyses using IL-1β[14,40–42]. Safranin-O staining of femoral heads cultured for 2 weeks with or without IL-1β displayed consistent decreases in proteoglycans for each genotype (Fig. 4a). Immunohistochemistry confirmed efficient knockdown of Runx2 in R2-Hetero and R2-Homo cKO femoral heads (Fig. 4a). Col2a1 protein levels were unchanged in normal cultures for both genotypes, but

decreased following IL-1β exposure, especially in the R2-Homo cKO group (Fig. 4a). qRT-PCR using mRNA from the homogenized femoral heads confirmed that *Runx2* was decreased according to genotype, but unchanged by IL-1β exposure (Fig. 4b). *Mmp13* increased in all genotypes following IL-1β exposure, and the expression pattern of *Mmp13* between genotypes was similar to that of *Runx2* (Fig. 4b). *Sox9* and *Col2a1* were decreased by IL-1β and unchanged between genotypes with a marked decrease in *Col2a1* in R2-Homo cKO femoral heads following IL-1β exposure (Fig. 4b).

We next examined temporal changes in mRNA levels of marker genes in chondrocytes from the four genotypes exposed to 1 ng/mL IL-1β from 0 to 8 h. The mRNA level of *Runx2* was efficiently decreased in R2-Hetero and R2-Homo cKO chondrocytes, and stable in each genotype following IL-1β exposure (Fig. 4c). *Mmp13* increased with time and was suppressed in R2-Hetero and R2-Homo cKO chondrocytes compared with their respective controls at each time point. *Sox9* gradually decreased, and there was no difference between genotypes at any point examined. *Col2a1* was significantly suppressed in R2-Homo cKO more than 24 h after IL-1β exposure (Fig. 4c).

### Genome-wide analysis of Runx2 and Runx3 association profiles in chondrocytes determined by ChIP-seq and RNA-seq

To investigate mechanisms underlying chondrocyte regulation by Runx3 and Runx2, we planned ChIP-seq with an anti-FLAG antibody. For Runx3, we prepared SFZ cells from WT mice transfected with the FLAG-tagged Runx3 expression vector. For the ChIP-seq, 21,730 raw peaks met the peak calling criterion, and ~50% of all peaks mapped to an interval between ±50 and 500 kb from the transcriptional start sites (TSS) (Fig. 5a). Genomic Regions Enrichment of Annotations Tool (GREAT) Gene Ontology (GO) analysis[51] indicated that the extracellular structure organization and collagen fibril organization terms were the most significantly enriched in the gene set (Fig. 5a). De novo motif analysis of the top 1000 specific peaks using MEME-ChIP[52] identified a previously predicted Runx motif, TG(T/C)GG(T/C) (Fig. 5b). ChIP-seq data around genes encoding *Prg4* and *Acan* showed several peaks for Runx3 binding (Supplementary Fig. 9a–d). Relative luciferase activities of the regions around *Prg4* and *Acan* were increased by Runx3 overexpression (Supplementary Fig. 9b, d). To further investigate the alterations of gene expression profiles by Runx3 knockout, we performed RNA-seq using SFZ and DZ cells from R3-cKO and R3-Cntl mice (Supplementary Fig. 10a–d and Supplementary Tables 1–4). Among 18 genes up- or downregulated by more than twofold in the R3-cKO SFZ cells, nine were the top one-third peak nearest genes as shown in the ChIP-seq, including *Prg4* and *Mmp9* (Fig. 5c).

For Runx2, we performed ChIP-seq in accordance with previous reports[12,53], using chondrocytes treated with vehicle control (primary chondrocytes) or exposed to 1 ng/mL IL-1β (inflamed chondrocytes) derived from *Runx2-FLAG* mice (Hojo H. et al.)[54]. For the ChIP-seq, 37,163 raw peaks in primary chondrocytes and 14,571 raw peaks in inflamed chondrocytes met the peak calling criteria (Fig. 5d). Peak distributions between primary chondrocytes and inflamed chondrocytes were similar. In both groups, a striking enrichment was observed around the TSS; ~24% of all peaks from two ChIP-seq data are

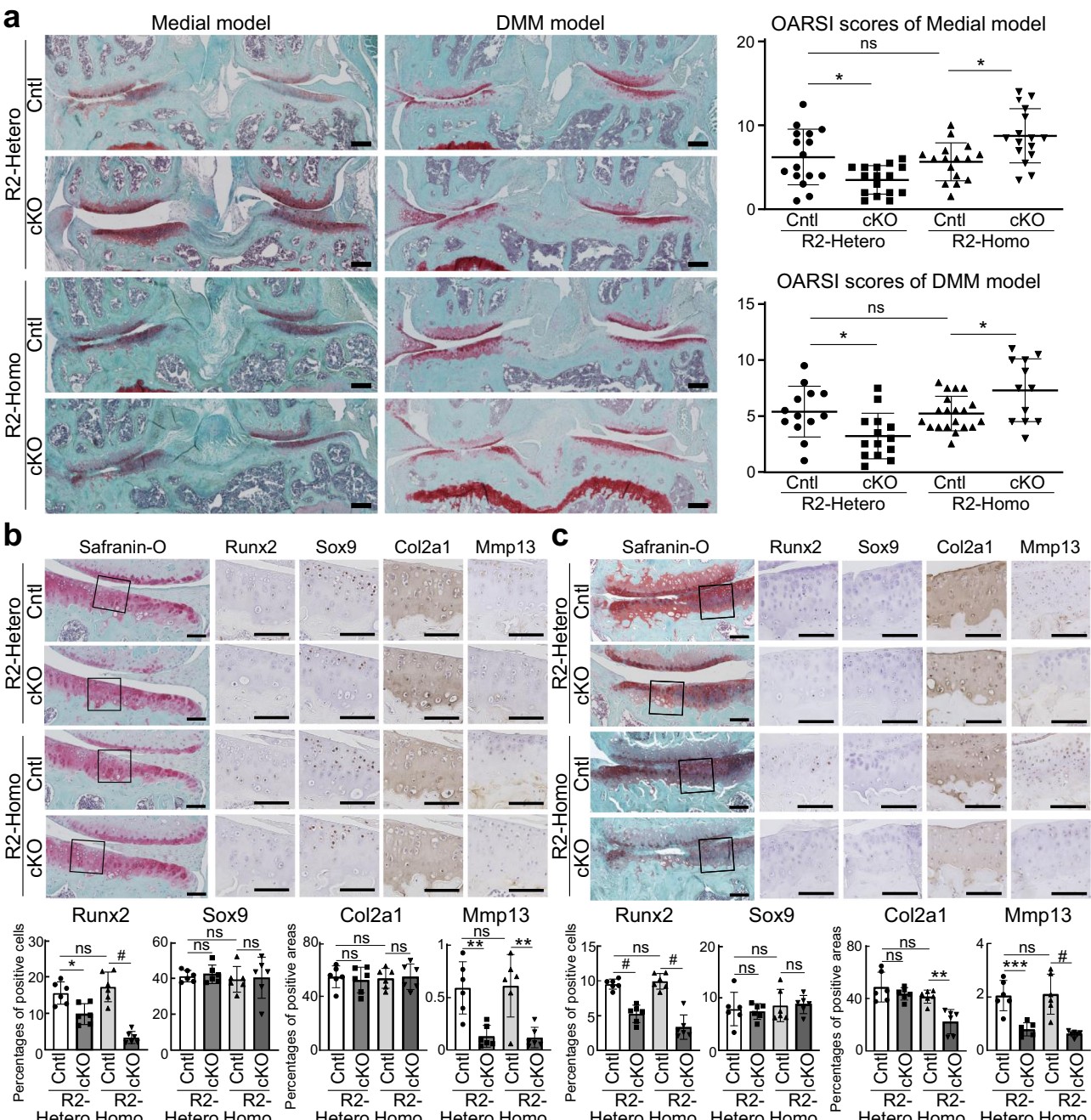

**Fig. 3 | Osteoarthritis development in chondrocyte-specific homozygous and heterozygous Runx2-knockout mice. a** Safranin-O staining and Osteoarthritis Research Society International (OARSI) scores of mouse knee joints of *Runx2*[fl/+] (R2-Hetero Cntl), *Col2a1-Cre*[ERT2]; *Runx2*[fl/+] (R2-Hetero cKO), *Runx2*[fl/fl] (R2-Homo Cntl), and *Col2a1-Cre*[ERT2]; *Runx2*[fl/fl] (R2-Homo cKO) mice 12 weeks after surgery to destabilize the medial meniscus (DMM) or 8 weeks after resection of the medial meniscus and medial collateral ligament (medial model). Tamoxifen induction was performed at 15 weeks in the DMM group and 7 weeks in the medial model group.

Experimental groups consisted of *n* = 13, 14, 20, and 12 in the DMM model, respectively, and *n* = 16 per genotype in the medial model. Scale bars, 200 μm. Safranin-O staining and immunohistochemistry of Runx2, Sox9, Col2a1, and Mmp13 in sham (**b**) and surgical (**c**) joints of the medial model group. *n* = 6 biologically independent experiments. Scale bars, 100 μm. Data are expressed as dot plots and mean ± SD. *P < 0.05, **P < 0.01, ***P < 0.001, #P < 0.0001; ns, not significant; one-way ANOVA test with Turkey's post hoc test.

within ±500 bp of the TSS, even though this region represents only 0.001% of the genome[51] (Fig. 5d). GREAT GO analysis[51] identified "collagen fibril organization" as the fifth and second most significantly enriched term in the gene sets for primary chondrocytes and inflamed chondrocytes, respectively (Fig. 5d). We hypothesized that there were functional differences between the TSS-associated dataset and dataset excluding data ±500 bp from the TSS, similar to Sox9[12]. GREAT GO analyses showed cell cycle-associated terms in TSS-associated Runx2

datasets of both primary and inflamed chondrocytes and terms for a Runx2-regulated skeletal program in the dataset excluding data ±500 bp from TSS (Supplementary Fig. 11a, b). A Venn diagram of GREAT GO analyses indicated that Runx2 was associated with collagen fibril organization in chondrocytes with or without inflammation (Fig. 5d).

We next performed de novo motif analysis of the top 1000 specific peaks using MEME-ChIP[52]. As expected, primary motifs were

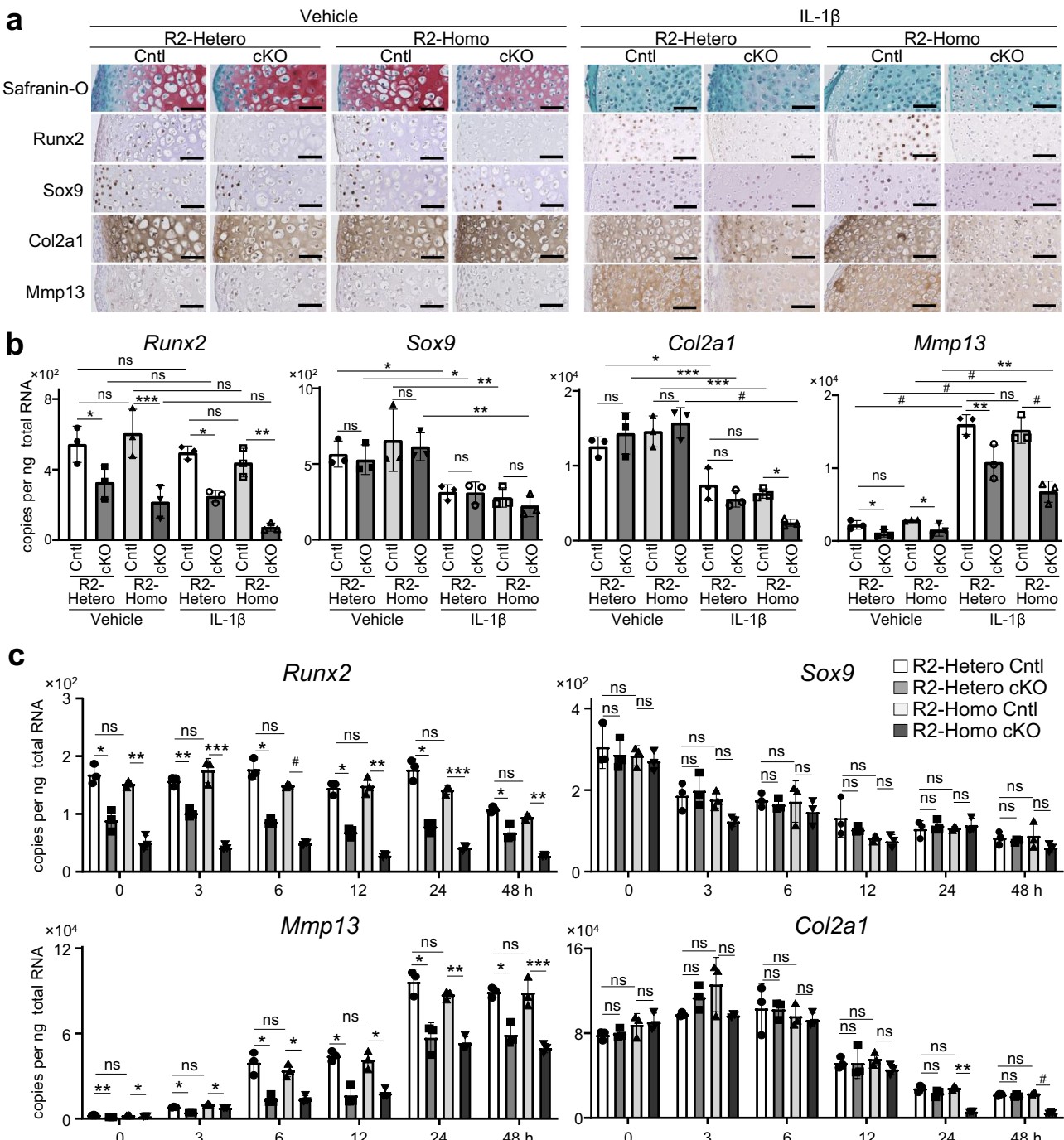

**Fig. 4 | Homozygous and heterozygous Runx2-knockout femoral heads and primary chondrocytes with or without interleukin-1-beta (IL-1β). a** Safranin-O staining and immunohistochemistry of Runx2, Sox9, Col2a1, and Mmp13 in femoral heads obtained from *Runx2*<sup>fl/+</sup> (R2-Hetero Cntl), *Col2a1-Cre*<sup>ERT2</sup>; *Runx2*<sup>fl/+</sup> (R2-Hetero cKO), *Runx2*<sup>fl/fl</sup> (R2-Homo Cntl), and *Col2a1-Cre*<sup>ERT2</sup>; *Runx2*<sup>fl/fl</sup> (R2-Homo cKO) P1 neonates, cultured with or without 1 ng per mL IL-1β for 2 weeks. Scale bars, 100 μm. **b** mRNA levels of *Runx2*, *Sox9*, *Col2a1*, and *Mmp13* in femoral heads.

**c** Time-course of *Runx2*, *Sox9*, *Col2a1*, and *Mmp13* mRNA levels in primary chondrocytes obtained from R2-Hetero Cntl, R2-Hetero cKO, R2-Homo Cntl, and R2-Homo cKO mice exposed to 1 ng per mL IL-1β. *n* = 3 biologically independent experiments. Data are expressed as dot plots and mean ± SD *$P < 0.05$, **$P < 0.01$, ***$P < 0.001$, #$P < 0.0001$; ns, not significant; one-way ANOVA test with Turkey's post hoc test.

similar to the previously predicted Runx motif, TG(T/C)GGT, even in the dataset excluding data ±500 bp from the TSS (Fig. 5e). Interestingly, we observed that poly-A sequence, reported as a SOX9 consensus sequences[55], was enriched in the TSS-associated Runx2 dataset. To identify the functional relationship between Sox9 and Runx2, we further analyzed our present dataset combined with previous Sox9 and related ChIP-seq datasets[12]. The TG(T/C)GGT motif was highly

enriched at the predicted center of Runx2 ChIP-seq peaks in primary and inflamed chondrocytes. Interestingly, the TG(T/C)GGT motif was slightly enriched at the predicted center of Sox9 ChIP-seq peaks (Fig. 5f). A poly-A sequence was also recovered, but there was no centering within the Runx2 and Sox9 peaks (Fig. 5f). Thus, the poly-A motif is unlikely to be the preferred primary site for Runx2 or Sox9 engagement in chondrocytes.

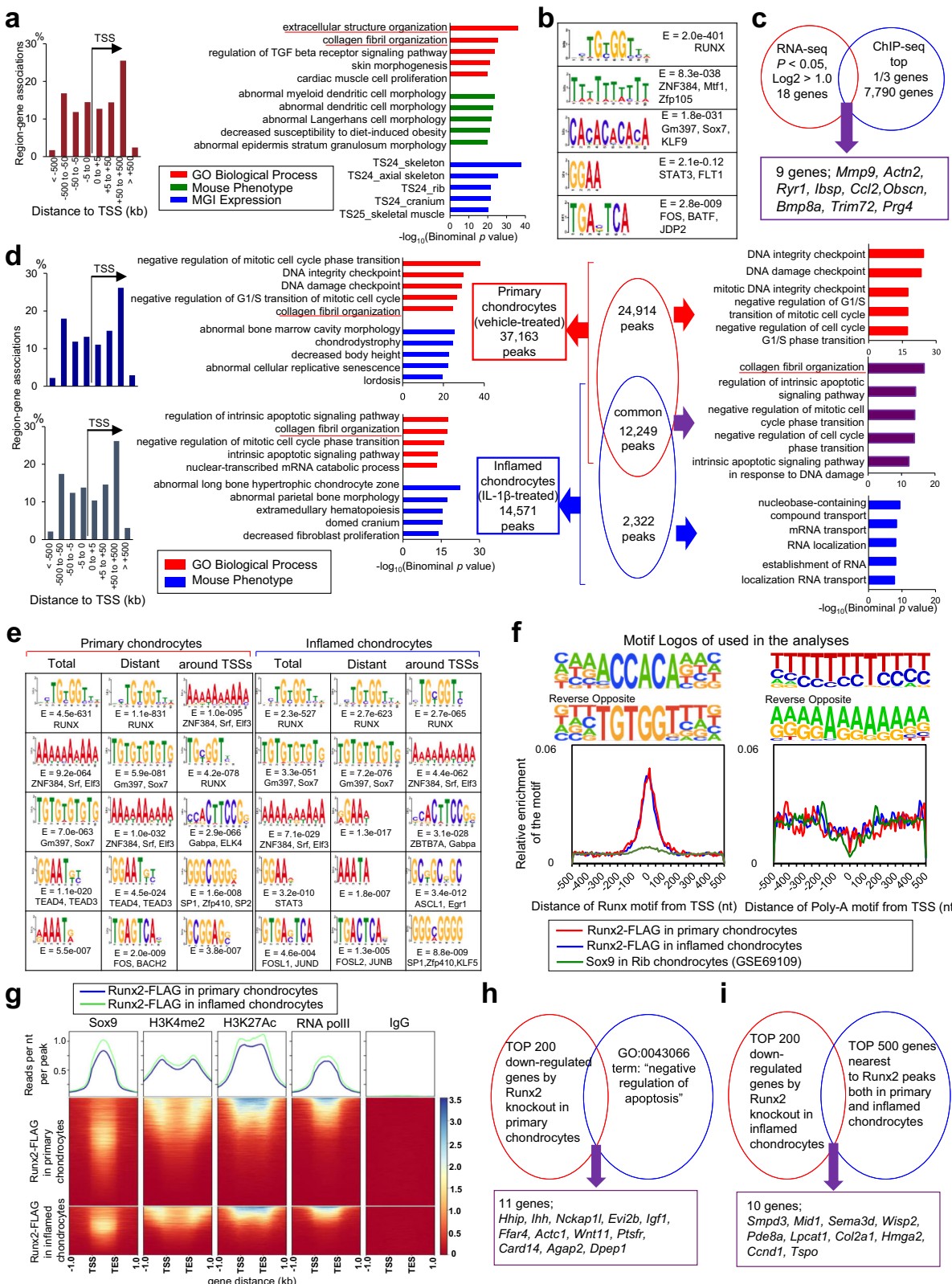

To test the integration of multiple regulatory inputs through Runx2- and Sox9-directed enhancer modules, we analyzed peak intensity associations. There were strong associations between Sox9-[12] and Runx2-peak regions, which were notably enhanced in inflamed chondrocytes (Fig. 5g). Clear associations of Runx2-peak regions with enhancer signatures were evident, specifically: (1) bi-modal patterns of H3K4 dimethylation (H3K4me2) peaks at Runx2-peak center regions, which indicates both promoters and putative enhancers;[56] (2) peaks of H3K27 acetylation (H3K27Ac) flanking Runx2 peaks, which indicate open chromatin[57,58], were more evident in inflamed chondrocytes compared with primary chondrocytes; and (3) RNA polymerase II peaks associated with Runx2-peak regions, consistent with active enhancers[59] (Fig. 5d). Together, these results indicate that Runx2 enhances transcription of collagen fibrils by binding TG(T/C)GGT

**Fig. 5 | Genome-wide analysis of Runx2 and Runx3 association profiles in chondrocytes. a** Genome-wide distribution of Runx3-associated regions relative to transcriptional start sites (TSS) with Genomic Regions Enrichment of Annotations Tool (GREAT) gene ontology (GO) and MGI expression annotations of Runx3 peaks showing the top five enriched terms. **b** Enriched motifs determined by Runx3-FLAG ChIP-seq. **c** Schematic diagram to identify candidate genes by ChIP-seq and RNA-seq using primary SFZ cells obtained from Cntl and cKO mice. **d** Genome-wide distribution of Runx2-associated regions relative to TSSs, and top five enriched terms identified by GREAT GO of Runx2 peaks. Left upper panel, primary chondrocytes treated with vehicle; left lower panel, inflamed chondrocytes exposed to 1 ng per mL IL-1β; right panels, overlap of Runx2-FLAG ChIP-seq peaks between primary and inflamed chondrocytes. **e** Enriched motifs determined by Runx2-FLAG ChIP-seq. **f** Enrichment of TG(T/C)GGT and poly-A motifs recovered de novo from

target sequences near Sox9-binding site, both in primary chondrocytes and inflamed chondrocytes.

To further investigate the general functions of Runx2 in chondrocytes, we performed RNA-seq analyses of R2-Homo Cntl and cKO chondrocytes with or without IL-1β exposure (Supplementary Fig. 12a–f and Supplementary Tables 5 and 6). The principal component analysis highlighted the effect of Runx2 knockout under inflammation induced by IL-1β. As chondrocyte apoptosis was enhanced in R2-Homo cKO mice (Supplementary Fig. 8b, c), we examined apoptosis-related genes. Among the top 200 genes downregulated by *Runx2* knockout in primary chondrocytes, eleven genes (including *Ihh*, *Igf1*, and *Wnt11*) were associated with "negative regulation of apoptosis" (Fig. 5h). Among the top 200 genes downregulated by Runx2 knockout in inflamed chondrocytes, ten (including *Col2a1*) matched with the top 500 genes nearest to Runx2 peaks detected by ChIP-seq (Fig. 5i).

## Transcriptional regulation of *Col2a1* and *Mmp13* by Runx2 and Sox9

We mapped Runx2-FLAG, Sox9[12], and active histone marks (H3K4me2 or H3K27Ac in chondrocytes[12] or osteoblasts[53] from 1-day-old mice) in ChIP-seq data using CisGenome browser. Sox9 and Runx2 sequence data showed similar patterns in introns 1 and 6 of *Col2a1*; specifically, insignificant peaks at intron 1 and major significant peaks at intron 6 (Fig. 6a). These peaks are identical to the functional enhancers of *Col2a1* containing Sox9 motifs[8]. Magnified views indicated that the peak centers of Runx2 and Sox9, containing their consensus motifs, were 300−400 bp apart (Supplementary Fig. 13). Based on the CisGenome browser, we prepared luciferase reporter vectors containing the Runx2 motifs around *Col2a1* and *Mmp13* genes shown in Fig. 6a. Relative luciferase activity of *Col2a1* intron 1 and 6 fragments decreased in chondrocytes exposed to ≥1 ng/mL IL-1β (Fig. 6b), which exhibited decreased *Sox9* mRNA while that of *Runx2* remained stable (Fig. 1c, d). However, luciferase activities of the region around the *Col2a1* TSS and upstream of the *Mmp13* TSS were unchanged compared with controls (Fig. 6b). We next examined the effect of Sox9 on the activity of each reporter. Sox9 overexpression increased the activities of reporters containing the enhancers in *Col2a1* introns 1 and 6 in a dose-dependent manner (Fig. 6c), but did not affect the activity of the region around the *Col2a1* TSS and decreased activity of the region upstream of the *Mmp13* TSS (Fig. 6c). In addition, we co-transfected Runx2 and Sox9 (Fig. 6d). Notably, when the amount of Sox9 was decreased, luciferase activities of the *Col2a1* intron 6 enhancer were increased by Runx2 co-transfection (Fig. 6d). In contrast, when Sox9 was overexpressed, luciferase activities of the *Col2a1* intron 1 enhancer were not affected by Runx2 transfection (Fig. 6d). Activity of the region around the *Col2a1* TSS was increased by Runx2 transfection without Sox9 transfection (Fig. 6d).

We further performed luciferase assays using chondrocytes from the four genotypes with or without IL-1β exposure. Relative luciferase activities of *Col2a1* intron 1 and 6 enhancers were decreased in

Runx2 and Sox9 peak regions. The upper panels show motif logos, displaying nucleotide frequencies (scaled relative to the information content) at each position. Lower panels show enrichment levels of TG(T/C)GGT and poly-A motifs mapped to Runx2 and Sox9 peaks, whereby x and y axes represent the distance from mapped motifs to the peak center and frequency of mapped motifs, respectively. **g** Signal intensity plots of ChIP-seq data for Sox9, H3K4me2, H3K27ac, RNA polymerase II (polII), and rabbit immunoglobulin G (IgG) controls in ±1 kb from Runx2-FLAG peaks. Normalized mean signal intensity for the following ChIP-seq reads are shown in the top panels. TES, transcriptional end site. **h** Schematic of identification of anti-apoptosis-associated candidates among downregulated genes by homozygous Runx2 knockout in primary chondrocytes. **i** Schematic of identification of candidate transcriptional target genes of Runx2 in inflamed chondrocytes.

inflamed chondrocytes compared with primary chondrocytes for each genotype (Fig. 6e), similar to the results shown in Fig. 6b. In addition, luciferase activities of *Col2a1* intron 1 and 6 enhancers, and the region around the *Col2a1* TSS, were suppressed in inflamed chondrocytes of R2-Homo cKO mice compared with other genotypes (Fig. 6e). Compared with controls, *Runx2* heterozygous or homozygous knockout decreased the activities of reporters containing the region upstream of the *Mmp13* TSS in both primary and inflamed chondrocytes (Fig. 6d). Taken together, the present results indicate that Runx2 could activate *Col2a1* transcription through the enhancer in intron 6, in addition to the enhancer in intron 1 and region around the *Col2a1* TSS.

## Enhanced expression of Runx3 inhibited surgically induced OA development

Finally, we investigated the effects of adenoviral Runx2 or Runx3 overexpression in vitro and in vivo. Transduced of a Runx2 adenovirus (Ad-Runx2) at different doses to WT chondrocytes increased *Runx2* expression in a dose-dependent manner (Fig. 7a). Although mRNA levels of *Sox9* and *Col2a1* were unchanged, *Mmp13* was upregulated in Runx2 overexpressing chondrocytes (Fig. 7a).

When we transduced Runx3 adenovirus at different doses to primary chondrocytes from WT mice, *Prg4* was increased in a dose-dependent manner (Fig. 7b). *Acan* and *Sox9* were also increased by Runx3 overexpression (Fig. 7b). Catabolic factors were not changed, except for slightly increased *Hif2a* (Fig. 7b). We next introduced GFP or Runx3 adenoviral vectors into knee joints of WT mice that received surgical induction for the medial model at 8 weeks of age. Runx3 expression was efficiently enhanced by the intra-articular injection of adenovirus, resulting in significant suppression of OA progression and enhanced Prg4 expression (Fig. 7c–f).

## Discussion

This study revealed that Runx3 protects articular cartilage degradation in mouse surgically induced OA models through extracellular matrix protein production, while Runx2 exerts both catabolic and anabolic effects under inflammatory conditions (Fig. 8). In vitro and comprehensive analyses of transcriptional regulation by Runx3 indicate its association with the organization of extracellular matrix, and identified *Prg4* in SFZ as a transcriptional target gene, as well as Acan in DZ as a potential target (Figs. 2 and 5). Consistent with previous studies, chondrocyte-specific heterozygous knockout of Runx2 inhibited OA and decreased *Mmp13* expression (Fig. 3). Notably, in contrast, homozygous knockout of *Runx2* in articular chondrocytes markedly accelerated OA development, which was accompanied by suppressed *Col2a1* protein level. Loss-of-function of Runx2 in vitro decreased *Col2a1* only in inflamed chondrocytes in which Sox9 was decreased (Fig. 4), while *Mmp13* expression was downregulated in both primary and inflamed chondrocytes. In the presence of IL-1β, de novo Runx2 binding to regions around the known Sox9-binding sites was enhanced (Fig. 5). Luciferase assays indicated that Runx2 can compensate for *Col2a1* transcription in inflamed chondrocytes (Fig. 6). Overexpression

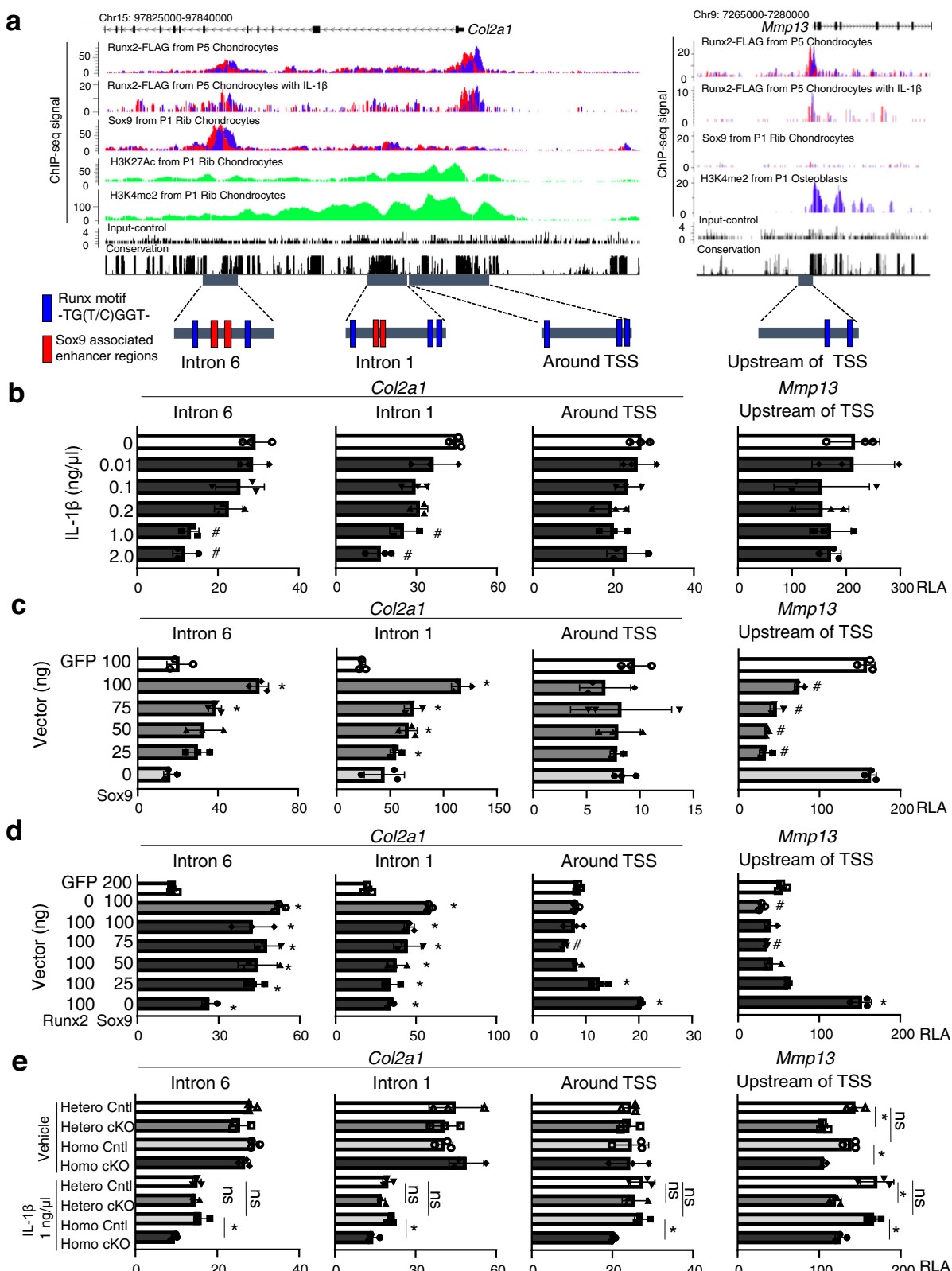

**Fig. 6 | Transcriptional regulation of *Col2a1* and *Mmp13* by Runx2 and Sox9.**
**a** CisGenome browser screenshots showing Runx2 and reference engagement around *Col2a1* (Chr15: 97,825,000 – 97,840,000) and *Mmp13* (Chr9: 7,265,000 – 7,280,000). Lower panels show genomic regions cloned into the pGL4.10[luc] vector. **b** Luciferase activities of enhancer regions in mouse chondrocytes exposed to various concentrations of IL-1β. **c, d** Luciferase activities of enhancer regions in mouse chondrocytes transfected with various ratios of Runx2 and Sox9, or GFP control. **e** Luciferase activities of enhancer regions in chondrocytes from *Runx2^{fl/+}* (R2-Hetero Cntl), *Col2a1-Cre^{ERT2}*; *Runx2^{fl/+}* (R2-Hetero cKO), *Runx2^{fl/fl}* (R2-Homo Cntl), and *Col2a1-Cre^{ERT2}*; *Runx2^{fl/fl}* (R2-Homo cKO) mice with or without IL-1β exposure. *n* = 3 (**b**–**e**) biologically independent experiments. Data shown mean ± SD. ns not significant. #*P* < 0.05 vs control (white bars) (**b**), **P* < 0.05 (**c**–**e**); one-way ANOVA test with Turkey's post hoc test. RLA relative luciferase activity.

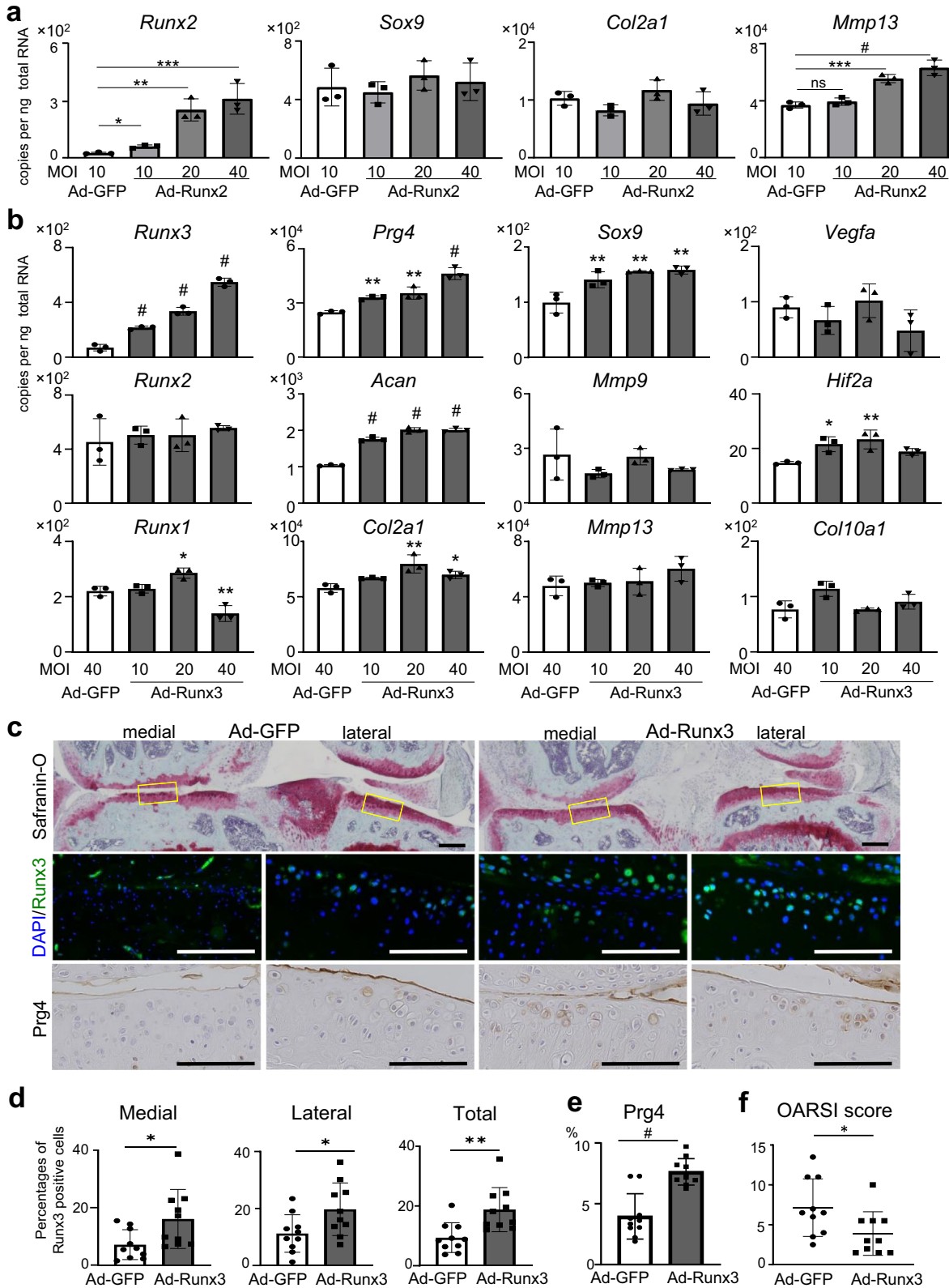

of Runx2 increased *Mmp13*, but not *Col2a1*. In contrast, overexpression of Runx3 increased *Prg4* and *Acan*, and intra-articular administration of Runx3 adenovirus suppressed surgically induced OA (Fig. 7).

In this study, OA development was enhanced in homozygous *Runx2*-knockout mice. This finding challenges the results of previous studies which directly or indirectly indicated catabolic effects of Runx2 in articular cartilage[31,32,60–62]. Compensation of *Col2a1* expression by

Runx2 was observed under inflammatory conditions in surgically induced OA models employing R2-Homo cKO mice, but not R2-Hetero cKO mice. This effect was not obvious in previous studies of general heterozygous-knockout mice[31] or *Aggrecan-Cre*[ERT2]; *Runx2*[fl/fl] mice in which the efficiency of *Runx2* knockdown was approximately 50%[32]. Considering the efficiency of *Runx2* suppression was ~80% in R2-Homo cKO mice, smaller amounts of Runx2 (compared with *Mmp13*

**Fig. 7 | Effects of Runx2 or Runx3 overexpression. a** mRNA levels of *Runx2*, *Sox9*, *Col2a1*, and *Mmp13* in WT primary chondrocytes overexpressing GFP (Ad-GFP) or Runx2 (Ad-Runx2) by adenoviral induction. **b** mRNA levels of Runx family members and marker genes in primary articular chondrocytes transduced with GFP (Ad-GFP) or Runx3 (Ad-Runx3) by adenoviral vectors. **c** Safranin-O staining and Runx3/Prg4 expression after OA development in WT knee joints after intra-articular administration of Ad-GFP or Ad-Runx3. Representative images among ten mice in each group were shown. All mice underwent medial model surgery at 8 weeks of age, and the intra-articular administration of 10 μL of 1 × 10⁹ plaque-forming unit per mL Ad- GFP or Ad-Runx3 was performed at 1, 4, and 7 weeks after the surgery. Inset boxes in Safranin-O staining indicate regions of immunofluorescence below. Scale bars indicate 200 μm. **d** Rates of Runx3-positive cells in medial, lateral, and both sides of knee joints. **e** Percentage of 3,3'-diaminobenzidine (DAB)-positive area in the Safranin-O-positive area. **f** OARSI scores for evaluation of OA development between the two groups. Data are expressed as dot plots and means ± SD of biologically dependent mice ($n = 3$ for **a**, **b**, $n = 10$ for **c**–**e**). *$P < 0.05$, **$P < 0.01$, ***$P < 0.001$, #$P < 0.0001$; ordinary ANOVA with Dunnett's multiple comparisons test for **a**, **b**, and two-tailed Mann–Whitney *U* test for **d**, **e**.

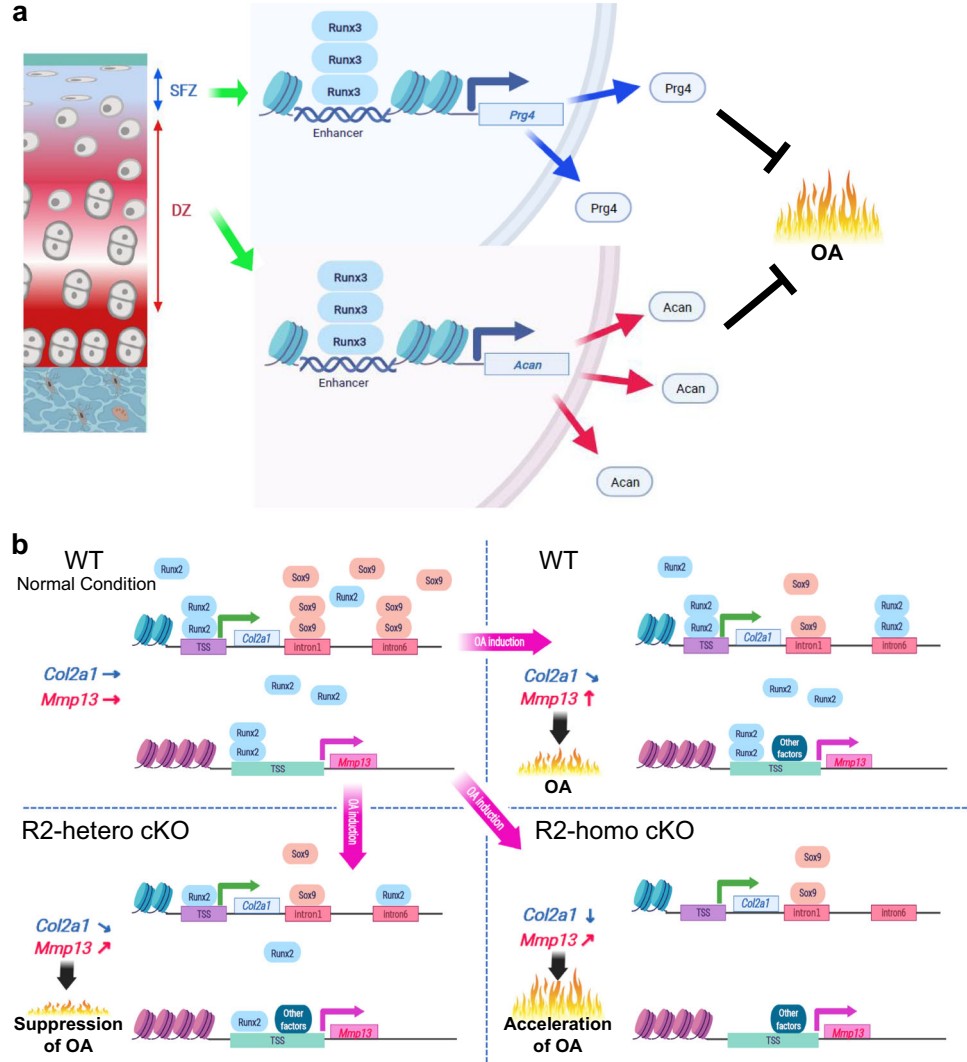

**Fig. 8 | Schematic diagrams representing molecular pathways in which Runx3 and Runx2 regulate articular cartilage during OA development. a** Runx3 contributes to cartilage homeostasis through the induction of Prg4 and Acan. **b** Runx2 exerts the catabolic effect through Mmp13 induction, while Runx2 sustains Col2a1 expression under the inflammatory condition, in which Sox9 was decreased.

transcription) may be able to sustain *Col2a1* transcription under inflammatory conditions. Although this anabolic effect of Runx2 is less prominent than its catabolic effect, it may contribute to the homeostasis of articular cartilage under pathological conditions in which Sox9 expression is decreased.

*Col2a1* expression was unchanged by *Runx2* homozygous knockout in primary chondrocytes, but decreased in inflamed chondrocytes (Fig. 4). These data were also supported by in vivo experiments showing that Col2a1 protein was not decreased in sham joints of R2-Homo cKO mice, but was significantly decreased in OA joints (Fig. 3d, e). A number of previous studies showed that Sox9 is most responsible for transcriptional induction of *Col2a1*[8,10–13], while Sox9 expression decreases during the early stage of OA development (Fig. 1). These data evoked our hypothesis that Runx2 may sustain *Col2a1* expression under inflammatory conditions where Sox9 is decreased. The results of ChIP-seq and luciferase assays showed that the enhancer in *Col2a1* intron 6, which contains both Sox9 and Runx2 consensus motifs, was activated by both transcription factors (Fig. 6c, d). The luciferase activity of the enhancer was tightly regulated by the amount of Sox9 and suppressed in accordance with decreased expression of Sox9 (Fig. 6c). Interestingly, although co-transfection of Runx2 and Sox9 did not enhance the activity of the

intron 6 enhancer, Runx2 significantly upregulated its activity when Sox9 was decreased (Fig. 6d). Accordingly, the intron 6 enhancer may be more important for transcriptional regulation of *Col2a1* by Sox9 and Runx2 than the intron 1 enhancer. Previous studies also showed that the intron 6 region is a functional enhancer for the *Col2a1* gene[8], and the transactivity of the intron 6 enhancer by Sox9 is stronger than that of the intron 1 enhancer[8,12]. All these data are consistent with our hypothesis, however, we could not further evaluate the biological effect sizes of Runx2/Sox9 binding to these enhancers. Considering that luciferase reporter assays often do not match in vivo data, it is necessary to perform OA model experiments using genome-edited mice in which these enhancers are mutated for further understanding.

In this study, we could not further clarify how Runx2 protein sustains *Col2a1* transcription when Sox9 protein decreases. A previous report displayed significant enrichment of TG(T/C)GGT motif in the top 2,000 Sox9 peaks in rib chondrocytes by ChIP-seq[12], which indicated (1) direct binding of Sox9 with the TG(T/C)GGT sequence, (2) its surrounding regions, or (3) indirect binding via interaction with Runx2. CisGeonme browser magnified view analysis supported the second idea, as Runx2 peak centers in *Col2a1* introns 1 and 6 are about 100 bp away from Sox9-binding sites[8,12] (Supplementary Fig. 13). In inflamed chondrocytes, de novo Runx2 binding to the Sox9-binding regions was enhanced (Fig. 5g). Under inflammatory conditions, decreased expression of endogenous Sox9 and subsequent decreased binding of Sox9 with its consensus motif may induce Runx2 to approach Runx motifs located near around the Sox9 motif. This hypothesis may also explain why Runx2 overexpression did not increase mRNA levels of *Col2a1* when endogenous Sox9 was sufficiently expressed (Fig. 7a). In contrast to Runx3 and Runx1, Runx2 may support cartilage anabolism only in pathogenic conditions.

ChIP-seq and RNA-seq results of this study displayed the different target genes of Runx2 and Runx3. However, because Runx family members have a common consensus motif, we could not identify the molecular mechanisms underlying how Runx2 and Runx3 distinguish their own target genes. Notably, expression patterns of these two proteins in articular cartilage during OA development are different (Fig. 1). Runx3 is expressed in all cartilage layers and decreased during OA development, whereas Runx2 expression is observed in the DZ and sustained during OA development (Fig. 1). These differences in expression are consistent with our hypothesis, but do not explain differences in their target gene specificity. Various factors, such as epigenomic regulation of genes and cofactors of each Runx protein, probably define transcriptional activation of Runx2- or Runx3-specific target genes.

In articular chondrocytes, Runx3 induces representative cartilage matrix gene, *Prg4*. Lubricin/Prg4, a mucin-like O-linked glycosylated protein, functions as a boundary lubricant in articular cartilage to decrease wear and friction, and the accumulation of lubricin at the surface of the cartilage is important for joint homeostasis[5,63]. In this study, both in vivo and in vitro experiments demonstrated that Prg4 is downregulated by Runx3 deficiency and Aggrecan/Acan, working as a shock absorber[64] and mainly expressed in DZ[65], is also a candidate of Runx3 transcriptional target (Figs. 2 and 5). In contrast to these anabolic effects, Runx3 deficiency did not change the expression of Mmp13 or proteoglycan release from articular cartilage (Fig. 2). Chondrocyte hypertrophy and apoptosis were also not changed by *Runx3* knockout (Supplementary Figs. 3 and 4) These data indicate that Runx3 does not directly affect the regulation of cartilage degradation or chondrocyte survival. ChIP-seq and RNA-seq results identified *Mmp9* as a target of Runx3 (Fig. 5c); however, consistent data for Mmp9 could not be obtained by qPCR and immunohistochemistry (Fig. 2). Although we did not examine other candidate genes, such as *Ccl* and *Ibsp*, they may play some roles in the regulation of articular cartilage as downstream molecules of Runx3.

*Mmp13* induction seems to be more straightforward. In the present study, *Mmp13* induction was consistently suppressed both by homozygous and heterozygous knockout of *Runx2* (Figs. 3 and 4), meanwhile, it was not affected by Runx3. These results indicate that Runx2 directly regulated *Mmp13* transcription, as indicated in previous reports[17–19,66,67]. Although Runx2 activated luciferase activity upstream of the *Mmp13* TSS, the Runx2 peak was not as large as we expected (Fig. 6). Obviously, *Mmp13* transcription is regulated by various signaling pathways and molecules other than Runx2. *Mmp13* was increased by IL-1β exposure in a dose- and time-dependent manner without increasing endogenous *Runx2* expression (Fig. 1c, d). We previously reported that NF-κB, HIF-2α, Notch signaling, and its downstream factor Hes1 are potent inducers of Mmp13[24,68,69]. Runx2 overexpression also indirectly induces MMP13 through mitogen-activated protein kinase pathways[17]. Considering that Mmp13 suppression by *Runx2* knockout was partial (Figs. 3 and 4), Runx2 must be associated with the induction of *Mmp13* transcription; however, its effect size remains unknown.

In conclusion, this study revealed that Runx3 protects articular cartilage against post-traumatic OA, while Runx2 exerts both anabolic and catabolic effects on articular cartilage under inflammatory conditions. Prg4 and Acan may be the main mediators of the anabolic effects of Runx3, whereas Runx3 does not affect cartilage catabolism. Runx2 induces *Mmp13*, whereas it compensates for reduced *Col2a1* expression in articular chondrocytes under inflammatory conditions in which Sox9 expression is decreased. Our findings regarding the anabolic roles of Runx3 and the diverse regulation of chondrocytes by Runx2 may contribute to understanding the pathophysiology of OA.

## Methods
### Animals
All animal experiments were authorized and approved by the Animal Care and Use Committee of The University of Tokyo. We have complied with all relevant ethical regulations. In each experiment, we compared the genotypes of littermates maintained in a C57BL/6J background. *Runx2*-floxed mice were generated to carry a conditional *Runx2* allele with exon 4, which encodes the Runt domain, flanked by loxP sites. Deletion of exon 4 also induced a frameshift and disrupted the Runx2 protein after exon 5. *Runx2^flox-neo* mice were crossed with flippase recombinase target (FRT) transgenic mice to eliminate the neomycin cassette. Positive embryonic stem cells were injected into eight-cell stage embryos to generate chimeric mice for subsequent crossing with C57BL/6 J females. This step provided heterozygous animals for the floxed allele of *Runx2* (*Runx2^flox-neo*). Mating these mice with FRT transgenic mice resulted in heterozygous-floxed *Runx2* (*Runx2^fl/+*) with the FRT-franked neomycin resistance cassette eliminated. *Runx2^fl/+* mice were next crossed with *Runx2^fl/+* to generate *Runx2^fl/+* or *Runx2^fl/f* mice. We then evaluated the targeting efficiency and specificity of these mice by breeding them with CAG-Cre reporter mice.

*Col2a1-Cre* mice[70], *Runx3^fl/fl* mice[46], and *Prg4-Cre^ERT2* mice[48] were purchased from Jackson Laboratory. *Col2a1-Cre^ERT2* mice[47] were generously provided by Professor Di Chen (Rush University Medical Center, Chicago, IL, USA). To generate *Col2a1-Cre; Runx3^fl/fl* mice, *Runx3^fl/fl* mice were mated with *Col2a1-Cre* mice to obtain *Col2a1-Cre; Runx3^fl/+* mice, which were then mated with *Runx3^fl/fl* mice. *Col2a1-Cre^ERT2; Runx3^fl/fl* mice and *Prg4-Cre^ERT2; Runx3^fl/fl* mice were created in the same manner. To generate *Col2a1-Cre^ERT2; Runx2^fl/fl* mice, *Runx2^fl/fl* mice were mated with *Col2a1-Cre^ERT2* mice to obtain *Col2a1-Cre^ERT2; Runx2^fl/+* mice, which were then mated with *Runx2^fl/fl* mice. After generating *Col2a1-Cre^ERT2; Runx2^fl/fl* mice, they were mated with *Runx2^fl/+* mice to obtain *Runx2^fl/+* (R2-Hetero Cntl), *Col2a1-Cre^ERT2; Runx2^fl/+* (R2-Hetero cKO), *Runx2^fl/fl* (R2-Homo Cntl), and *Col2a1-Cre^ERT2; Runx2^fl/fl* (R2-Homo cKO) mice, which were used in experiments.

*Runx2-Biotin-3xFLAG*-knockin mice (Runx2-FLAG mice), in which a Biotin-3×FLAG tag was inserted into the C-terminus of *Runx2* to produce Runx2-Biotin-3×FLAG fusion protein, were generated by one of the co-authors (Hojo, et al.[54]). Primer sequences for genotyping are shown in Supplementary Table 7.

## OA experiment

We used three models for OA experiments. For the medial model[41], tamoxifen (100 μg per gram of body weight) was intraperitoneally injected into 7-week-old male mice daily for 5 days, and surgery was performed at 8 weeks of age. Surgery was then performed at 8 weeks of age. Under general anesthesia, resection of the medial collateral ligament and medial meniscus was performed using a surgical microscope[41]. For the DMM model[45], tamoxifen (100 μg per gram of body weight; Sigma-Aldrich, St Louis, MO, USA) was intraperitoneally injected into 15-week-old male mice daily for 5 days, and surgery was performed at 16 weeks of age, in accordance with a previous report[45]. Mice were analyzed 8 weeks and 12 weeks after surgery for *Runx3*-knockout mice or *Runx2*-knockout mice, respectively. For the aging model, tamoxifen was injected for 5 days at 2, 6, and 12 months, and mice were raised until 18 months of age under physiological conditions. All mice were maintained under identical conditions (three mice per cage). OA severity was quantified by the Osteoarthritis Research Society International (OARSI) system[71], which was assessed by two observers blinded to the experimental groups.

## Histological analysis

The perfusion fixation was performed soon after euthanasia. Tissue samples were fixed with 4% paraformaldehyde in phosphate-buffered saline (PBS, pH 7.4) at 4 °C for 1 day. Specimens were decalcified with 10% ethylenediaminetetraacetic acid (EDTA, pH 7.4) at 4 °C for 2 weeks, embedded in paraffin, and cut into 4-μm-thick sagittal sections. Safranin-O staining was performed according to standard protocols. For immunohistochemistry, sections were incubated with antibodies against Runx3 (ab135248, Abcam, Cambridge, UK), Runx2 (ab192256, Abcam), Runx1 (ab23980, Abcam), Col2a1 (MAB8887, Merck, Darmstadt, Germany), Col10a1 (14-9771-80, Invitrogen, Carlsbad, CA, USA), Acan (13880-1-AP, Proteintech, Rosemont, IL, USA), Prg4 (ab28484, Abcam), Mmp13 (18165-1-AP, Proteintech), Mmp9 (ab38898, Abcam), and Sox9 (ab185230, Abcam). For visualization, 3,3'-diaminobenzidine (DAB; Nakaraitesk, Kyoto, Japan) was used. TdT-mediated dUTP nick-end labeling (TUNEL) staining was performed with an In Situ Cell Death Detection Kit (Roche, Basel, Switzerland) according to the manufacturer's instructions. Histological analyses were performed at least three times using 3−5 mice per group or genotype for confirmation of results. Images were visualized under a fluorescence microscope (BZ-X710, Keyence, Osaka, Japan). Percentages of positive cells and DAB-positive areas were measured by BZ analyzer software (Keyence) and NDPscan (Hamamatsu Photonics, Hamamatsu, Japan), respectively. The following parameters were used in the software: Pixel Area = 4.86E-08, Hue Value = 0.1, Hue Weight = 0.5, and Color Saturation Threshold = 0.04. We defined the strong intensity threshold as 0 – 100, medium as 100−175, and weak as 175−220 in the algorithm input. After margining the Safranin-O-positive articular cartilage area of lateral and medial sides of tibial and femoral joints by referring to the nearest slide, we evaluated areas strongly positive for Col2a1, and medium to strongly positive areas for Mmp13, respectively. We also counted numbers of Runx2/Sox9 medium/strong-positive cells. For quantification, we selected six samples exhibiting average OA development in each group.

## Cell cultures

Primary articular chondrocytes were isolated from 5-day-old C57BL/6J mice according to a standard protocol using collagenase D (Roche, Basel, Switzerland)[45], and cultured in Dulbecco's Modified Eagle's Medium (DMEM; Wako, Osaka, Japan) with 10% fetal bovine serum. The medium was changed every 3 days. SFZ cells were isolated as previously described[72]. Briefly, the proximal end of the femur and the distal end of the tibia were dissected from 5-day-old mice, and ligaments and tendons were excised. Cartilage tissues were incubated with 0.25% trypsin (Thermo Fisher Scientific, Waltham, MA, USA) for 1 h, followed by 1.5-h digestion with 173 U/mL of type I collagenase (Worthington Biochemical Corporation, Lakewood, NJ, USA). DZ cells were isolated by additional digestion of residual epiphyseal cartilage tissue with 43 U/ml collagenase type I for 5 h. Dissociated cells were seeded on fibronectin-coated culture dishes. Cells were cultured with DMEM with 10% FBS. The cells were cultured as a monolayer in all experiments. To induce Cre recombination in cultured cells, 2 μM 4-hydroxytamoxifen (Sigma-Aldrich) was applied for 48 h. IL-1β was added 24 h after starting this incubation.

## Explant cultures

For ex vivo cultures, femoral head cartilage was removed from Hetero Cntl, Hetero cKO, Homo Cntl, and Homo cKO mice at postnatal day 1. Femoral heads were then cultured in DMEM containing 10% fetal bovine serum, with or without 1 ng/μL IL-1β (AF-200-01B, PeproTech, Rocky Hill, NJ, USA). The medium was changed every 3 days, and the total culture period was 2 weeks. Right femoral heads were used for histological analyses and left ones were homogenized using Trizol reagent (15596-018, Life Technologies, Carlsbad, CA, USA) for qRT-PCR.

## qRT-PCR

Total RNA was purified with an RNeasy Mini Kit (Qiagen, Hilden, Germany). One microgram of total RNA was reverse transcribed using ReverTraAce qPCR RT Master Mix with gDNA Remover (Toyobo, Osaka, Japan). Each PCR reaction contained 1 × THUNDERBIRD SYBR qPCR Mix (Toyobo), 0.3 mM specific primers, and 25 ng of cDNA. The copy number was normalized to rodent total RNA (Thermo-Fisher Scientific, Waltham, MA, USA), with rodent glyceraldehyde-3-phosphate dehydrogenase (*Gapdh*) used as an internal control. All reactions were run in triplicate. Primer sequences are shown in Supplementary Table 8.

## Construction of expression vectors

Full-length cDNA sequences were amplified by PCR and cloned into the pCMV-HA or pShuttle vector (Clontech). Adenovirus vectors were generated using an Adeno-X expression system (Clontech). All vectors were verified by DNA sequencing.

## Luciferase assay

Col2a1 enhancer sites (upstream of TSS, intron 1, and intron 6), upstream of the Mmp13 TSS, Prg4, and Acan were amplified by PCR using mouse genomic DNA as the template and cloned into the PGL4.10 [luc2] vector (Promega, Tokyo, Japan). A luciferase assay was performed using primary chondrocytes and the Dual-Luciferase Reporter Assay System (Promega). Results are reported as the ratio of firefly to Renilla activities. Locations of the cloned sequences are shown in Supplementary Table 9.

## 1, 9-dimethylmethylene blue (DMMB) assay

The GAG content was measured using a 1, 9-dimethylmethylene blue (DMMB) assay[50]. Lysates (20 μL) were mixed with 200 μL of a DMMB (Sigma-Aldrich) working solution for 30 min at room temperature. The absorbance was then measured at 525 nm. Chondroitin sulfate (Sigma-Aldrich) was used as a standard.

## ChIP-seq

Chromatin preparation and ChIP were performed as previously described[9,53,73], with minor modifications. Briefly, $5 \times 10^7$ primary chondrocytes from Runx2-FLAG mice, expressing 3 × FLAG-tagged

Runx2, were cultured for 48 h with or without 1 ng/mL IL-1β for 24 h. For Runx3, we used SFZ chondrocytes from WT mice that were treated by lipofection (Lipofectamine 2000; Thermo Fisher Scientific) with vectors expressing 3xFLAG-tagged Runx3. Briefly, $5 \times 10^7$ chondrocytes were harvested and cross-linked with 1% formaldehyde, followed by quenching with glycine. After cell lysis, chromatin was fragmented by sonication (Branson Digital Sonifier 250, Branson Ultrasonics Corporation Danbury, CT, USA) to obtain 100–600 bp DNA fragments. The fragmented DNA lysate was incubated with Dynabeads M-280 sheep anti-mouse (Thermo Fisher Scientific) and antibody complex overnight at 4 °C [anti-FLAG Clone M2 (Sigma-Aldrich) or Ant-mouse IgG (Promega, Madison, WI, USA)]. After washing the beads with radioimmunoprecipitation assay buffer, DNA was eluted with elution buffer at 65 °C for 15 min, and the elution was incubated overnight at 65 °C for reverse crosslinking. DNA was purified using ChIP DNA Clean and Concentrator (ZYMO Research, Irvine, CA, USA). All buffers contained Complete EDTA-Free Protease Inhibitor Cocktail (Roche). Construction of ChIP-seq libraries was performed with a ThruPLEXFD Prep Kit (Takara Bio, Kusatsu, Japan). The library was sequenced on a HiSeq2000 (Illumina, San Diego, CA, USA) platform.

Supercomputing resources were provided by the XXX. DNA sequence information was aligned to the unmasked mouse genome reference sequence mm9 by a bowtie aligner[74]. Peak calling was performed by two-sample analysis using CisGenome software v2.0 (35) with a *P* value cut-off of $1 \times 10^{-5}$ compared with the input control. Peaks with a false discovery rate of <0.01 were incorporated into further analysis. GREAT GO analysis was performed utilizing the online GREAT GO program, version 3.0.0[52]. Each peak category was run against a whole-genome background with assembly mm9. De novo motif recovery was performed with MEME-ChIP[52]. A 100-bp region surrounding the peak center was extracted from mm9 and used for analyses. Raw and processed data are available in the Gene Expression Omnibus database under accession numbers "GSE157328" and "GSE165086". Peak data were compared with "GSM1692996", "GSM1692997", "GSM1693001", "GSM1693003", and "GSM1693005"[11].

## RNA-seq

For Runx3, RNA samples were collected from SFZ/DZ chondrocytes of 5-day-old *Col2a1-Cre; Runx3^fl/fl* and *Runx3^fl/fl* mice (*n* = 3, respectively). For Runx2, RNA samples were collected from primary chondrocytes of 5-day-old *Col2a1-Cre^ERT2^; Runx2^fl/fl* and *Runx2^fl/fl* mice (*n* = 2, respectively) and then subjected to 48-h incubation with 4-hydroxytamoxifen with or without 1 ng/μL IL-1β for 24 h. RNA-seq was conducted on a BGIseq-500 (BGI, Hong Kong, China) to produce 100-bp paired-end reads, which were mapped to the mouse genome (mm9). Quantile normalization was applied, and read counts were transformed to log2 counts per million. Linear models were used to test for expression differences between deficient and control samples. Raw and processed data are available in the Gene Expression Omnibus database under accession numbers "GSE157433" and "GSE165084".

## Intra-articular administration of adenoviral vectors

We performed intra-articular administration of an adenoviral vector containing Runx3 or GFP. Injections were performed three times: 1, 4, and 7 weeks after the OA surgery. For each administration, we injected 10 μL of $1 \times 10^9$ plaque-forming units of adenoviral vectors solution diluted in PBS.

## Statistics and reproducibility

Results were analyzed using Prism version 8.4.1 (GraphPad Software, San Diego, CA, USA). Two-tailed Mann–Whitney *U* test was used to establish statistical significance between the two groups. One-way ANOVA with Turkey's post hoc test was used to establish statistical significance in more than three groups. Ordinary one-way ANOVA with

Dunnett's multiple comparisons test was used for dose-dependent or time-course comparisons of multiple groups. *P* values below 0.05 were considered significant. Biologically independent sample numbers were shown in figures and figure legends. For micrographs, representative images in each group were displayed.

## Reporting summary

Further information on research design is available in the Nature Research Reporting Summary linked to this article.

## Data availability

Raw and processed data are available in the Gene Expression Omnibus database under accession numbers "GSE157328" and "GSE165086" for ChIP-seq. Peak data of "GSM1692996", "GSM1692997", "GSM1693001", "GSM1693003", and "GSM1693005" were used from the previous report[12]. "GSE157433" and "GSE165084" for RNA-seq. Source data are provided with this paper.

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

## Acknowledgements

We thank J. Sugita, R. Homma, A. Ogikubo, and K. Kaneko for technical assistance. We thank Ashleigh Cooper, PhD, from Edanz (https://jp.edanz.com/ac) for editing a draft of this manuscript. This work was supported by JSPS KAKENHI Grant Numbers 19H05654, 19H05565, 18KK0254, 17H04310, 20H03799, 20K09428, and 20K09451.

## Author contributions

Study conception and design: K.N., H.H., S.H.C., and T.S. Data collection: K.N., S.H.C., D.M., Yuma M., M.K., T.K., Y.I., H.N., and Yuji M. Data analysis: K.N., H.H., H.O., R.C., and S.O. Data interpretation: K.N., F.Y., R.C., Y.O., N.T., H.I., J.H., Y.T., U.C., S.T., and T.S. Generation of Runx2-cKO mice: T.S. Generation of Runx2-FLAG mice: H.H., S.O., and U.C. Drafting of the paper: K.N. and T.S. All authors proofread and approved the paper.

## Competing interests

The authors declare no competing interests.
