## [Peer Review File · Nature Communications]

REVIEWER COMMENTS

Reviewer #1 (Remarks to the Author):

The authors investigated the functions of Runx2 and Runx3 in OA by surgical induction using conditional KO mice. They found that Runx3 has anabolic functions by inducing Prg4 and Acan, while Runx2 has anabolic and catabolic functions by sustaining Col2a1 expression but inducing Mmp13 expression. Further, they showed that Runx2 sustained Col2a1 expression through an intron 6 enhancer when Sox9 expression was reduced in inflammatory condition, and that intra-articular administration of Runx3 protected the OA development. The findings are interesting and contribute to the understanding of the pathogenesis of OA with inflammation. However, some data are not convincing. Scientific comments are found below.

1. Immunohistochemistry is important in this paper, because it will be difficult to get enough RNA from OA cartilage. However, the data of immunohistochemistry are not convincing. For example, immunohistochemistry of Acan and Mmp13 seems to be not working, because the staining is not clear in the pictures.

2. In Fig. 1b, Runx2 seems to be expressed in SFZ.

3. The intensities of Col2a1 were quite different among the sections. The intensity deeply depends on each sample preparation especially in fixation. The fixation seems to have been done by immersion but not vascular perfusion. It is not easy to get constant staining by immunohistochemistry using the samples by immersion fixation. Is the measurement of the intensity of Col2a1 staining really quantitative?

4. Runx2 was detected in DZ but not in SFZ, but Col2a1 was reduced in the whole layer in the articular cartilage of R2-Homo cKO mice in OA model. Please check the staining again.

5. In Fig. 2F, Col10a1 in R3-Control was stained strongly in SFZ.

6. The expression and function of Runx3 in growth plate cartilage and articular cartilage seem to be quite different. Please discuss the reason.

7. Explain the reason for the increase of apoptosis in sham R2-Homo KO mice but not in the OA models.

8. Show the full spelling of TES in the legend of Fig. 5g.

9. In Supplementary Fig. 9a, c, how the peaks in Prg4 and Acan were selected? The detailed information of ChIP seq of Prg4 and Acan should be shown as Supplementary Fig. 11.

10. In Fig. 6a, Runx2-FLAG peak is not apparent in intron 1. Is there a significant peak?

11. If authors can show the Sox9-FLAG from P5 chondrocytes with or without IL-1b in Fig. 6a. It will be more understandable.

12. In Fig. 6a, there are no seq conservation in Sox9 and upstream Runx2 motif regions in intron 6, and in the downstream two Runx motif regions in intron 1 of Col2a1 gene. Further, there is no seq conservation in the upstream Runx motif region in Mmp13 gene. Are these sites really required for the enhancer activity?

13. Is the upregulation of the reporter activity by Sox9 and Runx2 enhanced in the reporter assay using intron 1 plus 6 in Fig. 6d?

14. As the results obtained from in vitro reporter assay do not match to the in vivo data frequently, the data from in vitro reporter assay of the enhancer candidates in Fig. 6 are suggestive but not conclusive.

15. ChIP-seq revealed that overlapping of Runx2 and Sox9 peaks was enhanced by exposure to IL-1 β (Fig. 5) (lines 372-373). What does it mean? Does Runx2 enhance the binding of Sox9 in inflamed chondrocytes?

16. When the knockout efficiency reached more than 50%, both Col2a1 and Mmp13 were suppressed, which was consistent with our results for R2-Homo cKO mice (Fig. 3, 4) (lines 419-421). As Col2a1 was not reduced in R2-Homo cKO mice, it is not consistent.

17. In line 435, Fig. 6c should be Fig. 6d.

18. Show the figure number in figures. It was very inconvenient.

Reviewer #3 (Remarks to the Author):

Comments to the Author

In this paper the authors explore the roles of Runx2 and Runx3 on chondrocytes during surgically induced arthritis. This subject has been well explored previously with a review of relevant literature on Runx2 discussed by Chen et al. (J Orth Transplant, 2019). Further, the results are predictable with Nishimura et al. (JBC, 2012) and Komori (Mol Cells, 2020) providing some potential explanation for some observed results in this study, especially the effects of Runx2 knockdown v. knockout and the effects of reduced cartilage. There is some novelty in the Runx3 component of the study, as prior work on Runx3 in arthritis has focused, almost entirely, on ankylosing spondylitis, where its loss is a risk factor. The literature was reasonably reviewed in the introduction, although Nishimura et al., with its link between Mmp13 and Runx2, wasn't cited directly. The general lack of novelty significantly reduces enthusiasm for publication.

The results are significant insofar as they are thorough and interesting in support of studying osteoarthritis, which is very interesting per se. However, the significance of the results is undermined by their substantial lack of novelty. The thoroughness of the results allows them to provide new and interesting angles on, e.g. Runx3, where the adenovirus induced overexpression of Runx3 results in therapeutic chondrogenic effects.

The approach is generally reasonable, although they didn't seem to have considered the novelty well enough during the development of their approach. Their choice of experiments, however, is reasonable and, as I'll discuss later, thorough in confirming prior results showing that OA is affected by Runx family genes, both Runx2 and Runx3.

Their results are very thorough, using a variety of methods both in vivo and in vitro to approach the question of Runx2 and Runx3's effects on osteoarthritis and chondrocytes. They focus on surgically induced osteoarthritis and safranin O (histology), standard methods for assessing osteoarthritis progression and cartilage health. Results are presented for histological, and immunohistochemical analysis after induction of osteoarthritis and during aging, as well as qPCR (Fig 1). They examine the effects of Runx3 knockout on OA histologically and with gene expression, including the effects of rescue using viral Runx3 overexpression (Fig 2). Likewise, they examine the effects of Runx2 heterozygous and homozygous knockout on OA models DMM and medial model with IHC and qPCR (Fig 3+4). They follow-up with ChIP-seq and RNA-seq results showing where Runx2 and Runx3 bind and gene expression changed in various conditions (Fig 5). They also looked at known targets of Runx2 and Sox9, such as Mmp13 (per Nishimura et al. (JBC, 2012)), Col2a1, and Il1b, using high throughput methods (Fig 6). They finally look at the use of adenovirus based Runx2/3 overexpression showing some changes in the expression of genes associated with better OA prognosis with Runx3 overexpression, along with some safranin O staining (Fig 7) – although the results here appear to contradict their claim in the abstract that, “Intra-articular administration of Runx3 adenovirus ameliorated development of surgically induced OA.”

The writing quality of the paper is sufficient for publication although some errors are apparent, e.g. line 798 on page 32 “mRNA levels of Ruxn2...”. There were a few errors, but generally the writing quality of the paper is good and simply needs some proofing.

Ultimately, this is a very thorough research paper covering results that are eminently predictable from prior work and/or already previously explored. The results are of a high quality, but are, unfortunately, not sufficiently novel for publication in Nature Communications.

Major concerns:

- The results of this paper are substantially similar to prior work, other issues are relatively minor.

Minor concerns:

- There are a few errors in the writing, e.g. line 798 on page 32 “mRNA levels of Ruxn2...”.

Reviewer #1 (Remarks to the Author):

The authors investigated the functions of Runx2 and Runx3 in OA by surgical induction using conditional KO mice. They found that Runx3 has anabolic functions by inducing Prg4 and Acan, while Runx2 has anabolic and catabolic functions by sustaining Col2a1 expression but inducing Mmp13 expression. Further, they showed that Runx2 sustained Col2a1 expression through an intron 6 enhancer when Sox9 expression was reduced in inflammatory condition, and that intra-articular administration of Runx3 protected the OA development. The findings are interesting and contribute to the understanding of the pathogenesis of OA with inflammation. However, some data are not convincing. Scientific comments are found below.

1. Immunohistochemistry is important in this paper, because it will be difficult to get enough RNA from OA cartilage. However, the data of immunohistochemistry are not convincing. For example, immunohistochemistry of Acan and Mmp13 seems to be not working, because the staining is not clear in the pictures.

We appreciate the careful review of our manuscript.

We agree with the reviewer's comments. We employed the same antibodies used in our recent studies for Acan (*J Bone Miner Metab.* 40:196–207, 2022) and Mmp13 (*Nat Commun.* 10:1442, 2019; *Arthritis Rheumatol.* 73:1441–50, 2021; *J Bone Miner Metab.* 40:196–207, 2022). However, because we carefully controlled the experimental conditions of IHC to suppress non-specific staining, the signal intensity in some images may be too weak. We have replaced IHC images, particularly Acan and Mmp13 in Figs. 1–4. With OA development, Mmp13 expression increases in the extracellular area of deep zone cartilage and subchondral bone, and is also detected in OA chondrocytes. Mmp13 was hardly detected in R3-Cntl, R3-cKO^{ER}, and R3-pKO^{ER} cartilage (Fig. 2b, d) because these were articular cartilage tissues of sham-operated knee joints, which displayed no obvious OA changes.

2. In Fig. 1b, Runx2 seems to be expressed in SFZ.

It was inappropriate that we displayed low-magnification images for Runx2 and Sox9 in Fig. 1b of the original manuscript. We have revised these images to show the same magnification as provided for Runx3 IHC. The revised Fig. 1b indicates Runx2 expression in the middle and deep zones.

3. The intensities of Col2a1 were quite different among the sections. The intensity deeply depends on each sample preparation especially in fixation. The fixation seems to have been done by immersion but not vascular perfusion. It is not easy to get constant staining by immunohistochemistry using the samples by immersion fixation. Is the measurement of the intensity

of Col2a1 staining really quantitative?

For all mice, we performed vascular perfusion fixation soon after euthanasia and fixed tissue samples by immersion for 1 day. We added an appropriate description in the “Histological analysis” section of the “Supplementary information”. In this section, we also described the procedure for quantification. Briefly, we classified the signal intensity into three categories (strong, medium, and weak) and defined “strong” as positive for Col2a1. In many studies, positive areas of IHC staining are quantified by similar procedures.

4. Runx2 was detected in DZ but not in SFZ, but Col2a1 was reduced in the whole layer in the articular cartilage of R2-Homo cKO mice in OA model. Please check the staining again.

We agree with the reviewer’s comment. We carefully checked the images of Col2 again. Col2 is detected in whole layers of normal articular cartilage, but the staining seems to be slightly more intense in the deep zone, i.e., under the tidemark. Meanwhile, the signal intensity of Col2 was decreased in whole layers of R2-Homo cKO cartilage (Fig. 3c). Although the observed decrease in Col2 in the deep layer is probably due to *Runx2* insufficiency, that in the superficial layer is not likely to be a direct effect of *Runx2* knockout. Other transcription factors for Col2a1, such as HIF-1alpha, Sox5, and Sox6, may be involved in the decrease of Col2 expression in the superficial layer. In the revised manuscript, we changed the representative image of Col2 in Fig. 3c to show a more obvious decrease of Col2 staining, particularly in the deep zone.

5. In Fig. 2F, Col10a1 in R3-Control was stained strongly in SFZ.

Thank you for pointing this out. Non-specific signals are often detected in the SFZ. Accordingly, we changed the representative image of Col10 staining in Fig. 2f of the revised manuscript.

6. The expression and function of Runx3 in growth plate cartilage and articular cartilage seem to be quite different. Please discuss the reason.

In Supplementary Fig. 4d, we displayed Runx3 expression in epiphyseal cartilage of an E18.5 embryo. The image shows Runx3 expression in prehypertrophic and hypertrophic chondrocytes, but not in the periarticular zone. We display another image of Runx3 IHC in the right panel, which we showed in our previous study (*Sci Rep.* 9:7666, 2019, Supplementary Fig. 5). Runx3 is expressed around the surface of the periarticular zone, similar to the expression pattern in adult articular cartilage.

7. Explain the reason for the increase of apoptosis in sham R2-Homo KO mice but not in the OA models.

During the terminal stage of OA (i.e., 8 weeks after medial model surgery and 12 weeks after the DMM surgery), evaluation by histological examination is often difficult because cartilage is severely degenerated by catabolic enzymes such as Mmp13 and Adamts5. In this stage, both the area of articular cartilage and number of chondrocytes were decreased. Decreases of cartilage area or chondrocyte number strongly affect quantitative data of immunohistochemical and TUNEL staining. For this reason, we generally used sham joints for immunohistochemistry. Because the TUNEL staining data of terminal OA joints did not show an accurate effect of Runx2 on apoptosis, we removed Supplementary Fig. 8a of the original manuscript to avoid confusion.

8. Show the full spelling of TES in the legend of Fig. 5g.

We have added this information to the figure legend.

9. In Supplementary Fig. 9a, c, how the peaks in Prg4 and Acan were selected? The detailed information of ChIP seq of Prg4 and Acan should be shown as Supplementary Fig. 11.

We described how to select peaks in the Supplementary Methods as below:

“Peak calling was performed by two-sample analysis using CisGenome software v2.0 (35) with a P-value cut-off of 1×10^{-5} compared with the input control. Peaks with a false discovery rate of < 0.01 were incorporated into further analysis.”

Regarding the latter comment, we have shown detailed information for ChIP-seq of *Prg4* and *Acan* in Supplementary Fig. 13 (we assume that the reviewer intended to refer to Supplementary Fig. 13, not Supplementary Fig. 11).

10. In Fig. 6a, Runx2-FLAG peak is not apparent in intron 1. Is there a significant peak?

The peaks of Runx2-FLAG in intron 1 with or without IL-1 β were not significant. We revised this description in the Results section to avoid misleading (line 309). Although the peaks in intron 1 were not significant, we examined the intron 1 element because it is a representative binding element for Sox9 (*Mol Cell Biol.* 17:2336–46, 1997; *J Biol Chem.* 273:14989–97, 1998).

11. If authors can show the Sox9-FLAG from P5 chondrocytes with or without IL-1b in Fig. 6a. It will be more understandable.

We appreciate the reviewer’s comment. We noticed a mistake in the description: we used previously acquired ChIP-seq data using an endogenous Sox9 antibody (*Cell Rep.* 12:229–43, 2015), not Sox9-FLAG mice. We have corrected this description in the revised manuscript.

We agree with the reviewer’s comment. We tried Sox9 ChIP-seq using primary mouse chondrocytes with IL-1 β . Regrettably, the ChIP-seq experiment did not succeed, probably because Sox9 expression levels decrease in chondrocytes under inflammatory conditions, as shown in Fig. 1. We added a description of this issue in the third paragraph of the Discussion section of the revised manuscript.

12. In Fig. 6a, there are no seq conservation in Sox9 and upstream Runx2 motif regions in intron 6, and in the downstream two Runx motif regions in intron 1 of Col2a1 gene. Further, there is no seq conservation in the upstream Runx motif region in Mmp13 gene. Are these sites really required for the enhancer activity?

First, we appreciate the reviewer’s comment. During the verification process, we found that the label for the cloned region indicating “Upstream of TSS” was incorrectly displayed in Fig. 6a. We revised the schematic of Fig. 6a to correctly state “Around TSS” because the cloned region also included the region downstream of TSS.

The above schematic is Fig. 6a, in which we numbered the elements from (1) to (14). The degree of conservation is as follows: (1) not conserved, (2) not conserved, (3) relatively conserved, (4) highly conserved, (5) highly conserved, (6) highly conserved, (7) highly conserved, (8) not conserved, (9) not conserved, (10) not conserved, (11) highly conserved, (12) highly conserved, (13) not conserved, and (14) highly conserved. Several studies comparing key selected mammals indicated rapid evolution of enhancers (*Cell* 149:1381–1392, 2012; *Cell* 154:185–196, 2013; *Cell* 160:554–66, 2015). Among them, Villar *et al.* performed comparative functional genomic analysis in 20 mammalian species and found that enhancers are rarely conserved across these mammals (*Cell* 160:554–66, 2015). Therefore, we believe that a low degree of conservation does not determine the biological significance of enhancers.

13. Is the upregulation of the reporter activity by Sox9 and Runx2 enhanced in the reporter assay using intron 1 plus 6 in Fig. 6d?

We agree with the reviewer that we could not determine the biological meaning of activation of these enhancers by Sox9 and Runx2 in independent experiments. Meanwhile, it is difficult to make a long construct containing from intron 1 to intron 6. It is possible to make a construct containing both enhancer regions by conjugating them; however, it would be difficult to interpret data from such an artificial enhancer sequence even though the reporter activity is enhanced. Although OA model experiments using genome-edited mice in which these enhancers are mutated would be very helpful to understand their significance, we could not perform such experiments in this study. We have described this issue in the third paragraph of the Discussion section of the revised manuscript.

14. As the results obtained from in vitro reporter assay do not match to the in vivo data frequently, the data from in vitro reporter assay of the enhancer candidates in Fig. 6 are suggestive but not conclusive.

We agree with the reviewer. Comprehensive analyses identified genes downregulated in inflamed chondrocytes by Runx2 knockout (Fig. 5i). Considering that Col2 is the most essential cartilage matrix protein and that *Col2a1* was included in these ten genes, the observed decrease of Col2 in R2-Homo cKO cartilage is likely associated with enhanced OA development in R2-Homo cKO mice. However, regarding the mechanism underlying compensation of *Col2a1* expression by Runx2 under inflammatory conditions, we could not provide other evidence in addition to the results of reporter assays. We have described this issue in the third paragraph of the Discussion section in the revised manuscript.

15. ChIP-seq revealed that overlapping of Runx2 and Sox9 peaks was enhanced by exposure to IL-1 β (Fig. 5) (lines 372-373). What does it mean? Does Runx2 enhance the binding of Sox9 in inflamed chondrocytes?

We have revised this description as follows: "In the presence of IL-1 β , *de novo* Runx2 binding to regions around the known Sox9-binding sites was enhanced" (lines 370-371).

16. When the knockout efficiency reached more than 50%, both *Col2a1* and *Mmp13* were suppressed, which was consistent with our results for R2-Homo cKO mice (Fig. 3, 4) (lines 419-421). As *Col2a1* was not reduced in R2-Homo cKO mice, it is not consistent.

As the reviewer pointed out, *Col2a1* expression was reduced only in OA cartilage of R2-Homo cKO mice, but not in their normal cartilage. Notably, the cited paper (reference number 34 in the original manuscript) reported effects of Runx2 in temporomandibular joints during skeletal growth (tamoxifen was injected at 5 weeks of age and mice were analyzed at 9 weeks), but not in adult

articular cartilage. To improve accuracy, we have substantially revised the description of this topic in the Discussion section of the revised manuscript.

17. In line 435, Fig. 6c should be Fig. 6d.

We deeply appreciate the reviewer's comment. We have corrected this error in the revised manuscript (line 403).

18. Show the figure number in figures. It was very inconvenient.

We apologize for the inconvenience. We were not aware that the figure number was not shown in the merged PDF and have corrected this error in the revised manuscript.

Reviewer #3 (Remarks to the Author):

Comments to the Author

In this paper the authors explore the roles of Runx2 and Runx3 on chondrocytes during surgically induced arthritis. This subject has been well explored previously with a review of relevant literature on Runx2 discussed by Chen et al. (*J Orth Transplant*, 2019). Further, the results are predictable with Nishimura et al. (*JBC*, 2012) and Komori (*Mol Cells*, 2020) providing some potential explanation for some observed results in this study, especially the effects of Runx2 knockdown v. knockout and the effects of reduced cartilage. There is some novelty in the Runx3 component of the study, as prior work on Runx3 in arthritis has focused, almost entirely, on ankylosing spondylitis, where its loss is a risk factor. The literature was reasonably reviewed in the introduction, although Nishimura et al., with its link between Mmp13 and Runx2, wasn't cited directly. The general lack of novelty significantly reduces enthusiasm for publication.

We appreciate the careful review of our manuscript.

As we described in the original manuscript, previous studies showed that OA development was suppressed by heterozygous Runx2 knockout, and Mmp13 was downregulated by Runx2 knockout. Nishimura *et al.* (*J Biochem*, 2012) displayed effects of osterix and Runx2 on hypertrophic differentiation and Mmp13 expression, and the aforementioned two reviews (Komori et al. *Mol Cells*, 2022 and Chen et al. *J Orth Transplant*, 2019) introduced the promotive effects to OA by Runx2. Taken together, all these previous studies showed the catabolic effects of Runx2 on OA.

On the basis of our previous findings revealing various effects of NF- κ B/RelA and canonical WNT pathway signaling in articular cartilage (*Nat Commun.* 10:1442, 2019; *Nat Commun.* 7:13336, 2016; *Arthritis Res Ther.* 21:247, 2019), we hypothesized that Runx2 may have roles other than its

catabolic effects exerted via Mmp13 induction. In our preliminary experiments, we found that Runx2 was expressed in the deep layer of normal articular cartilage, which lead us to comprehensively investigate roles of Runx2.

In this study, we found that Runx2 contributes to the maintenance of articular cartilage under inflammatory conditions by supporting Col2a1 transcription. We only displayed amelioration of OA development with Mmp13 suppression in Runx2 hetero-cKO mice (as shown by previous studies) to compare the phenotypes of Runx2 hetero- and homo-cKO. Our novel findings identifying anabolic roles of Runx2 and Runx3 are useful for understanding OA pathophysiology and developing OA therapeutics. Although many researchers claim that Runx2 is a therapeutic target because of its catabolic effects on cartilage, the anabolic role of Runx2 may also be worth considering.

To make this point clearer, we have revised the Introduction section. In addition, two review papers (Komori et al. *Mol Cells*, 2022 and Chen et al. *J Orth Transplant*, 2019) and the article (Nishimura et al. *J Biochem*, 2012) have been cited in the Introduction and Discussion sections of the revised manuscript.

The results are significant insofar as they are thorough and interesting in support of studying osteoarthritis, which is very interesting per se. However, the significance of the results is undermined by their substantial lack of novelty. The thoroughness of the results allows them to provide new and interesting angles on, e.g. Runx3, where the adenovirus induced overexpression of Runx3 results in therapeutic chondrogenic effects.

The approach is generally reasonable, although they didn't seem to have considered the novelty well enough during the development of their approach. Their choice of experiments, however, is reasonable and, as I'll discuss later, thorough in confirming prior results showing that OA is affected by Runx family genes, both Runx2 and Runx3.

We appreciate the reviewer's comments. We performed all possible experiments, including heterozygous knockout of Runx2, to confirm the previously reported catabolic effects of Runx2.

Their results are very thorough, using a variety of methods both in vivo and in vitro to approach the question of Runx2 and Runx3's effects on osteoarthritis and chondrocytes. They focus on surgically induced osteoarthritis and safranin O (histology), standard methods for assessing osteoarthritis progression and cartilage health. Results are presented for histological, and immunohistochemical analysis after induction of osteoarthritis and during aging, as well as qPCR (Fig 1). They examine the effects of Runx3 knockout on OA histologically and with gene expression, including the effects of rescue using viral Runx3 overexpression (Fig 2). Likewise, they examine the effects of Runx2 heterozygous and homozygous knockout on OA models DMM and medial model with IHC and qPCR (Fig 3+4). They follow-up with ChIP-seq and RNA-seq results showing where Runx2 and

Runx3 bind and gene expression changed in various conditions (Fig 5). They also looked at known targets of Runx2 and Sox9, such as Mmp13 (per Nishimura et al. (JBC, 2012)), Col2a1, and Il1b, using high throughput methods (Fig 6). They finally look at the use of adenovirus based Runx2/3 overexpression showing some changes in the expression of genes associated with better OA prognosis with Runx3 overexpression, along with some safranin O staining (Fig 7) – although the results here appear to contradict their claim in the abstract that, “Intra-articular administration of Runx3 adenovirus ameliorated development of surgically induced OA.”

We appreciate the reviewer’s comments. The results shown in Figure 7 do not contradict our claim in the abstract. Overexpression of Runx3 enhanced expression of *Prg4*, *Sox9*, *Acan*, and *Col2a1*, but did not increase *Mmp13* (Fig. 7b). Accordingly, intra-articular administration of Runx3 adenovirus suppressed OA development. We confirmed enhanced expression of *Mmp13* by Runx2 overexpression *in vitro*, but did not perform Runx2 overexpression *in vivo*.

The writing quality of the paper is sufficient for publication although some errors are apparent, e.g. line 798 on page 32 “mRNA levels of Ruxn2...”. There were a few errors, but generally the writing quality of the paper is good and simply needs some proofing.

Ultimately, this is a very thorough research paper covering results that are eminently predictable from prior work and/or already previously explored. The results are of a high quality, but are, unfortunately, not sufficiently novel for publication in Nature Communications.

Major concerns:

- The results of this paper are substantially similar to prior work, other issues are relatively minor.

As described above, we aimed to investigate the complete roles of Runx2 and Runx3 in adult articular cartilage, and revealed novel anabolic effects and their mechanisms. We have revised the manuscript to more clearly state the novelty of our findings.

Minor concerns:

- There are a few errors in the writing, e.g. line 798 on page 32 “mRNA levels of Ruxn2...”.

We appreciate the comment. We have corrected this error in the revised manuscript.

REVIEWERS' COMMENTS

Reviewer #1 (Remarks to the Author):

The authors responded to my concerns appropriately except my first comment. In Fig. 2a-d, the manuscript and the legend are wrong.

Manuscript:

As a result of the medial model, OA development was significantly accelerated in R3-cKOER (Fig. 2a, b) and R3-pKOER (Fig. 2c, d) knee joints compared with R3-Cntl joints. In the articular cartilage of sham-operated R3-cKOER joints, Runx3-positive cells decreased by 70% (Fig 2b).

Legend:

Development of OA in Runx3fl/fl (R3-Cntl) and Col2a1-CreERT2;Runx3fl/fl (R3-cKOER) mice (a, b) and in R3-Cntl and Prg4-CreERT2;Runx3fl/fl (R3-pKOER) mice (c, d) by the medial model,

They showed Fig. 2b and d as OA models as well as sham-operated mice. OARSI scores should be 2b and 2d.

Reviewer #2 (Remarks to the Author):

Comments to the Author

In this paper the authors explore the roles of Runx2 and Runx3 on chondrocytes and OA during surgically induced arthritis. Although some of the findings are of interest, the study is lack of novelty. The results of the study are highly predictable from prior work and already previously explored.

Major concerns:

1. The role of Runx2 OA has been studied extensively by other labs as shown some of publications listed below and the study is lack of novelty. The results of the study are highly predictable from prior work and already previously explored.

Runx2 plays a central role in Osteoarthritis development. PMID: 32913706 Free PMC article. Review. Chondrocyte-Specific RUNX2 Overexpression Accelerates Post-traumatic Osteoarthritis Progression in Adult Mice. . PMID: 31189030 Free PMC article.

miR 204 5p inhibits the occurrence and development of osteoarthritis by targeting Runx2. PMID: 30106092 Free PMC article.

Identification of a novel, methylation-dependent, RUNX2 regulatory region associated with osteoarthritis risk. PMID: 30010910 Free PMC article.

miR-105/Runx2 axis mediates FGF2-induced ADAMTS expression in osteoarthritis cartilage. PMID: 26816250

Deletion of Runx2 in Articular Chondrocytes Decelerates the Progression of DMM-Induced Osteoarthritis in Adult Mice. PMID: 28539595 Free PMC article.

WNT16 is upregulated early in mouse TMJ osteoarthritis and protects fibrochondrocytes against IL-1 β induced inflammatory response by regulation of RUNX2/MMP13 cascade. PMID: 33301961

Downregulation of microRNA-29b by DNMT3B decelerates chondrocyte apoptosis and the progression of osteoarthritis via PTHLH/CDK4/RUNX2 axis. PMID: 33177241 Free PMC article.

Association of GDF5, SMAD3 and RUNX2 polymorphisms with temporomandibular joint osteoarthritis in female Han Chinese. PMID: 25757091

Col10a1-Runx2 transgenic mice with delayed chondrocyte maturation are less susceptible to developing osteoarthritis. PMID: 25628784 Free PMC article.

MicroRNA-320a protects against osteoarthritis cartilage degeneration by regulating the expressions of

BMI-1 and RUNX2 in chondrocytes. PMID: 29441992

The NAD-Dependent Deacetylase Sirtuin-1 Regulates the Expression of Osteogenic Transcriptional Activator Runt-Related Transcription Factor 2 (Runx2) and Production of Matrix Metalloproteinase (MMP)-13 in Chondrocytes in Osteoarthritis. PMID: 27367673 Free PMC article.

Chondrocyte-intrinsic Smad3 represses Runx2-inducible matrix metalloproteinase 13 expression to maintain articular cartilage and prevent osteoarthritis. PMID: 22674505 Free PMC article.

Pomegranate extract inhibits the interleukin-1 β -induced activation of MKK-3, p38 α -MAPK and transcription factor RUNX-2 in human osteoarthritis chondrocytes. PMID: 20955562 Free PMC article.

2. Authors did not explained well for why "Runx2 conditional knockout mice showed biphasic phenotypes: heterozygous knockout inhibited OA and decreased matrix metalloproteinase (Mmp13) expression, while homozygous knockout of Runx2 accelerated OA and reduced type II collagen (Col2a1) expression", the data is not very convincing.

3. Phenotype osteoarthritis in Runx3-knockout mice is very mild, which indicates that Role of Runx3 deficiency in osteoarthritis may be not essential.

4. The study is lack of novelty and insight of molecular mechanism at the roles of Runx2 and Runx3 deficiency in OA.

Minor concerns:

- Page 2, line 21, "under the normal condition" is stilted and should be replaced with "under normal conditions"

- Page 3, line 52-53, "multiple upstream molecules of Mmp13" should be re-phrased as "multiple transcription factors upstream of Mmp13 expression"

- Page 4, line 59, "ossification" appears to be a misspelling of "ossification"

- Page 4, line 62, "OA development was significantly and expression..." appears to be an incomplete thought, where OA development should be significantly reduced. It's citing Kamekuma et al. (Arth Rheum, 2006).

- Supplementary information, page 1, line 7, mentions IACUC approval by "University of XXX" – please address this, as "XXX" doesn't sound like the name of the university.

Reviewer #1 (Remarks to the Author):

The authors responded to my concerns appropriately except my first comment.

In Fig. 2a-d, the manuscript and the legend are wrong.

Manuscript:

As a result of the medial model, OA development was significantly accelerated in R3-cKOER (Fig. 2a, b) and R3-pKOER (Fig. 2c, d) knee joints compared with R3-Cntl joints. In the articular cartilage of sham-operated R3-cKOER joints, Runx3-positive cells decreased by 70% (Fig 2b).

Legend:

Development of OA in Runx3fl/fl (R3-Cntl) and Col2a1-CreERT2;Runx3fl/fl (R3-cKOER) mice (a, b) and in R3-Cntl and Prg4-CreERT2;Runx3fl/fl (R3-pKOER) mice (c, d) by the medial model,

They showed Fig. 2b and d as OA models as well as sham-operated mice. OARSI scores should be 2b and 2d.

We are sorry that the figures and legends were confusing. Fig.2a and 2c are safranin O staining images and OARSI scores of surgically-induced OA knee joints, while Fig. 2b and 2d are IHC images of sham-operated knee joints. We could not accurately compare expression of marker proteins in OA knee joints between two mice groups with different OA development, because articular cartilage was degenerated and cartilage area was decreased in OA joints. Additionally, Runx3 is involved only in the early stage of OA. For these reasons, we used sham-operated knee joints for IHC to examine Runx3 effects on expression of these molecules. We have revised the descriptions in the manuscript (lines 120-125) and legend.

Reviewer #2 (Remarks to the Author):

Comments to the Author

In this paper the authors explore the roles of Runx2 and Runx3 on chondrocytes and OA during surgically induced arthritis. Although some of the findings are of interest, the study is lack of novelty. The results of the study are highly predictable from prior work and already previously explored.

Major concerns:

1. The role of Runx2 OA has been studied extensively by other labs as shown some of publications listed below and the study is lack of novelty. The results of the study are highly predictable from prior

work and already previously explored.

We appreciate the introduction of these papers. We have read all, but none of them indicated anabolic effects of Runx2 under inflammation during OA development, which we think the novelty of this paper. We have cited some of these papers in the Discussion (lines 370-371).

Runx2 plays a central role in Osteoarthritis development. PMID: 32913706 Free PMC article. Review. Chondrocyte-Specific RUNX2 Overexpression Accelerates Post-traumatic Osteoarthritis Progression in Adult Mice. . PMID: 31189030 Free PMC article.

miR 204 5p inhibits the occurrence and development of osteoarthritis by targeting Runx2. PMID: 30106092 Free PMC article.

Identification of a novel, methylation-dependent, RUNX2 regulatory region associated with osteoarthritis risk. PMID: 30010910 Free PMC article.

miR-105/Runx2 axis mediates FGF2-induced ADAMTS expression in osteoarthritis cartilage. PMID: 26816250

Deletion of Runx2 in Articular Chondrocytes Decelerates the Progression of DMM-Induced Osteoarthritis in Adult Mice. PMID: 28539595 Free PMC article.

WNT16 is upregulated early in mouse TMJ osteoarthritis and protects fibrochondrocytes against IL-1 β induced inflammatory response by regulation of RUNX2/MMP13 cascade. PMID: 33301961

Downregulation of microRNA-29b by DNMT3B decelerates chondrocyte apoptosis and the progression of osteoarthritis via PTHLH/CDK4/RUNX2 axis. PMID: 33177241 Free PMC article.

Association of GDF5, SMAD3 and RUNX2 polymorphisms with temporomandibular joint osteoarthritis in female Han Chinese. PMID: 25757091

Col10a1-Runx2 transgenic mice with delayed chondrocyte maturation are less susceptible to developing osteoarthritis. PMID: 25628784 Free PMC article.

MicroRNA-320a protects against osteoarthritis cartilage degeneration by regulating the expressions of BMI-1 and RUNX2 in chondrocytes. PMID: 29441992

The NAD-Dependent Deacetylase Sirtuin-1 Regulates the Expression of Osteogenic Transcriptional Activator Runt-Related Transcription Factor 2 (Runx2) and Production of Matrix Metalloproteinase (MMP)-13 in Chondrocytes in Osteoarthritis. PMID: 27367673 Free PMC article.

Chondrocyte-intrinsic Smad3 represses Runx2-inducible matrix metalloproteinase 13 expression to maintain articular cartilage and prevent osteoarthritis. PMID: 22674505 Free PMC article.

Pomegranate extract inhibits the interleukin-1 β -induced activation of MKK-3, p38 α -MAPK and transcription factor RUNX-2 in human osteoarthritis chondrocytes. PMID: 20955562 Free PMC article.

2. Authors did not explained well for why “Runx2 conditional knockout mice showed biphasic

phenotypes: heterozygous knockout inhibited OA and decreased matrix metalloproteinase (Mmp13) expression, while homozygous knockout of Runx2 accelerated OA and reduced type II collagen (Col2a1) expression”, the data is not very convincing.

We also do not know why Runx2 plays such diverse roles in articular cartilage. However, the *in vivo* data have shown that heterozygous and homozygous knockout of Runx2 lead to opposite phenotypes in OA development, and the mechanistic analyses indicate the anabolic effect of Runx2 under inflammation, in addition to its catabolic effects.

3. Phenotype osteoarthritis in Runx3-knockout mice is very mild, which indicates that Role of Runx3 deficiency in osteoarthritis may be not essential

We agree with the reviewer. Runx3 is involved only in the early stage of OA.

4. The study is lack of novelty and insight of molecular mechanism at the roles of Runx2 and Runx3 deficiency in OA.

We agree with the reviewer that the present data have not completely revealed all mechanisms underlying articular cartilage regulation by Runx2 and Runx3. However, we believe that all comprehensive analyses in the present study, including histological examinations, *in vitro* experiments, RNA-seq, ChIP-seq, and promoter analyses, explain how Runx2 and Runx3 contribute to cartilage homeostasis and OA development.

Minor concerns:

- Page 2, line 21, “under the normal condition” is stilted and should be replaced with “under normal conditions”

- Page 3, line 52-53, “multiple upstream molecules of Mmp13” should be re-phrased as “multiple transcription factors upstream of Mmp13 expression”

- Page 4, line 59, “ossification” appears to be a misspelling of “ossification”

- Page 4, line 62, “OA development was significantly and expression...” appears to be an incomplete thought, where OA development should be significantly reduced. It’s citing Kamekuma et al. (Arth Rheum, 2006).

We appreciate the careful review of our revised manuscript. We have corrected the descriptions mentioned above.

- Supplementary information, page 1, line 7, mentions IACUC approval by “University of XXX” – please address this, as “XXX” doesn’t sound like the name of the university.

We deleted the name of our institute, because of double-blind peer review. We will open the specific name when the manuscript is accepted.